# Visual recognition of social signals by a tectothalamic neural circuit

Johannes M. Kappel[1], Dominique Förster[1], Katja Slangewal[1,4], Inbal Shainer[1], Fabian Svara[1,2], Joseph C. Donovan[1], Shachar Sherman[1], Michał Januszewski[3], Herwig Baier[1✉] & Johannes Larsch[1✉]

Social affiliation emerges from individual-level behavioural rules that are driven by conspecific signals[1–5]. Long-distance attraction and short-distance repulsion, for example, are rules that jointly set a preferred interanimal distance in swarms[6–8]. However, little is known about their perceptual mechanisms and executive neural circuits[3]. Here we trace the neuronal response to self-like biological motion[9,10], a visual trigger for affiliation in developing zebrafish[2,11]. Unbiased activity mapping and targeted volumetric two-photon calcium imaging revealed 21 activity hotspots distributed throughout the brain as well as clustered biological-motion-tuned neurons in a multimodal, socially activated nucleus of the dorsal thalamus. Individual dorsal thalamus neurons encode local acceleration of visual stimuli mimicking typical fish kinetics but are insensitive to global or continuous motion. Electron microscopic reconstruction of dorsal thalamus neurons revealed synaptic input from the optic tectum and projections into hypothalamic areas with conserved social function[12–14]. Ablation of the optic tectum or dorsal thalamus selectively disrupted social attraction without affecting short-distance repulsion. This tectothalamic pathway thus serves visual recognition of conspecifics, and dissociates neuronal control of attraction from repulsion during social affiliation, revealing a circuit underpinning collective behaviour.

Many animals live in groups, the result of a basic social affiliative drive, which requires detection and approach of conspecifics. Social affiliation is a prerequisite of consummatory actions such as aggression, mating or play[3], and is also a proximal cause of swarm, flock and herd formation. Although neural circuits that mediate such behaviours have received much attention[3,15], relatively little is known about the sensory detection of social signals (beyond pheromones)[15,16], and how such cues feed into the regulation of social distance. One important class of visual social signals is biological motion, which comprises conspecific movement patterns that trigger complex approach and pursuit behaviours[17–19], and elicit a social percept in humans[9,10]. Biological motion is also a key driver of zebrafish shoaling, a collective behaviour with well-characterized interaction rules in groups or pairs of animals[2,6,11,20], offering a model to investigate visual neural circuits underpinning social affiliation.

## Neuronal activity during social affiliation

To identify the relevant neural circuits in juvenile zebrafish aged 21 days, we generated unbiased maps of recent neuronal activity after shoaling with real or virtual conspecifics (Fig. 1a). Virtual conspecifics were projected black dots moving either with fish-like biological motion, or continuously, which are highly attractive and weakly

attractive, respectively[2] (Fig. 1b). We then recorded a snapshot of neuronal activity by rapid fixation and labelling of *c-fos* (official gene symbol, *fosab*) mRNA[21] using third-generation in situ hybridization chain reaction[22] (HCR) analysis in the forebrain, midbrain and anterior hindbrain (Fig. 1a,c and Extended Data Fig. 1). Visual inspection of the registered and merged *c-fos* signal from all of the animals identified 31 distinct clusters with robust activity in response to one or more stimulus conditions (Fig. 1c and Extended Data Fig. 1).

Splitting the data by stimulus group revealed that social context differentially activated these clusters. Activation by real and virtual conspecifics overlapped in a subset of clusters, including the lateral rostral hypothalamus (cluster Hrl) and intermediate hypothalamus (cluster Hi3), while showing a distinct pattern in other areas. In the optic tectum (TeO), virtual conspecifics activated a ventrolateral cluster, matching the retinotopic representation of the ventrally projected black dot visual stimulus[23]. By contrast, real conspecifics moving at the same elevation as the imaged fish activated the anterior and dorsal TeO more strongly (Fig. 1d). Virtual and real conspecifics activated a cluster in the dorsal thalamus (DT), and real conspecifics also activated an anterior cluster in the ventral thalamus (VT) (Fig. 1d). The DT *c-fos* cluster overlapped with an expression hotspot of the gene *cortistatin* (*cort*, also known as somatostatin 7 (*sst7*)), which we co-labelled using a third, multiplexed HCR probe and used subsequently as a DT landmark[24]

[1]Max Planck Institute for Biological Intelligence (formerly Max Planck Institute of Neurobiology), Planegg, Germany. [2]Max Planck Institute for Neurobiology of Behavior – caesar, Bonn, Germany. [3]Google Research, Zurich, Switzerland. [4]Present address: Centre for the Advanced Study of Collective Behaviour, University of Konstanz, Konstanz, Germany. ✉e-mail: herwig.baier@bi.mpg.de; johannes.larsch@bi.mpg.de

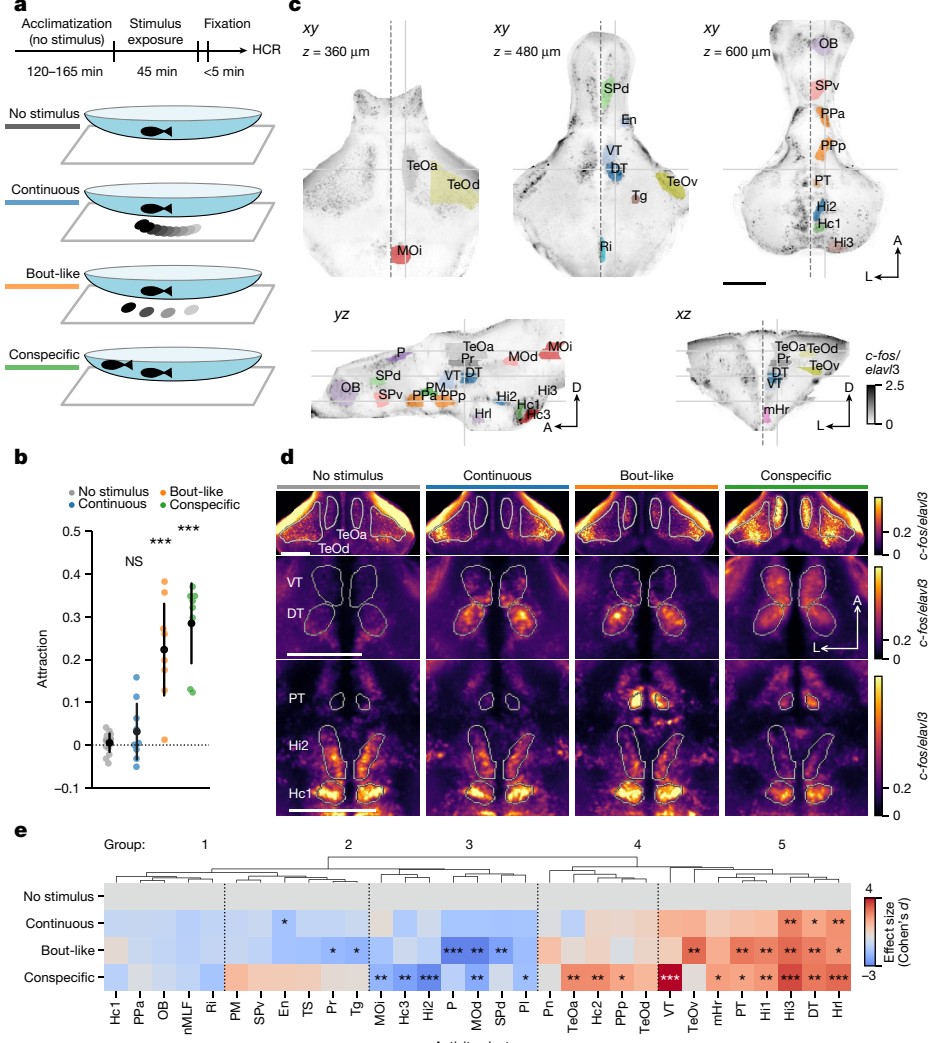

**Fig. 1 | Fish-like motion activates a conserved social behaviour network.**
**a**, Schematic of stimulus presentation for activity mapping. **b**, Attraction
towards stimuli shown in **a**. $n = 17$ (no stimulus) or $n = 9$ (continuous; bout-like)
single animals; and $n = 8$ animals tested in 4 pairs (conspecific). Data are mean
(black dots) ±1 s.d. Exact $P$ values were calculated using two-tailed $t$-tests
compared with the no-stimulus group: $P = 0.16$ (continuous); $P = 5.2 \times 10^{-8}$
(bout-like); $P = 8.1 \times 10^{-11}$ (conspecific). Bonferroni-corrected $\alpha$ values: NS,
$P > 0.05/3$ (NS); ***$P < 0.001/3$. **c**, Representative slices of maximum-intensity-
normalized $c$-$fos$ signal merged across all 28 registered animals. The views are
horizontal (top row), sagittal (bottom left) and coronal (bottom right). The solid
grey lines indicate the corresponding planes across the slices. The dashed line
indicates the midline. Coloured patches indicate activity clusters (Extended
Data Fig. 1). A, anterior; D, dorsal; L, lateral. **d**, The average normalized $c$-$fos$
signal at the three representative horizontal planes indicated in **c**. $n = 6$ (no
stimulus), $n = 8$ (continuous), $n = 6$ (bout-like) and $n = 8$ (conspecific) animals.
**e**, The effect size (Cohen's $d$) of normalized bulk $c$-$fos$ induction compared with
the no-stimulus condition. Negative values indicate a lower signal compared
with the no-stimulus condition. The dendrogram represents hierarchical

clustering. Statistical analysis was performed using two-tailed $t$-tests in each
activity cluster versus the no-stimulus group. *$P < 0.05/3$, **$P < 0.01/3$,
***$P < 0.001/3$ ($\alpha$ values were Bonferroni-corrected per activity cluster). Animal
numbers are the same as in **d**. Additional statistical information is provided as
Source Data. Scale bars, 200 µm. DT, dorsal thalamus; En, entopeduncular
nucleus; Hc1, caudal hypothalamus 1; Hc2, caudal hypothalamus 2; Hc3, caudal
hypothalamus 3; Hi1, intermediate hypothalamus 1; Hi2, intermediate
hypothalamus 2; Hi3, intermediate hypothalamus 3; Hrl, rostral hypothalamus,
lateral; mHr, rostral hypothalamus, medial; MOd, medulla oblongata, dorsal;
MOi, medulla oblongata, intermediate; nMLF, nucleus of the medial
longitudinal fasciculus; OB, olfactory bulb; P, pallium; Pl, pallium, lateral; PM,
magnocellular preoptic nucleus; Pn, pineal; PPa, anterior parvocellular preoptic
nucleus; PPp, posterior parvocellular preoptic nucleus; Pr, pretectum; PT,
posterior tuberculum; Ri, inferior raphe; SPd, subpallium, dorsal; SPv,
subpallium, ventral; TeOa, tectum, anterior; TeOd, tectum, dorsal; TeOv,
tectum, ventral; Tg, lateral tegmentum; TS, torus semicircularis; VT, ventral
thalamus.

(Extended Data Fig. 2a,b). One cluster in the posterior tuberculum (PT)
stood out as selectively active with bout-like motion and real conspecif-
ics. By contrast, $c$-$fos$ in the intermediate hypothalamic cluster Hi2 was
highest for the no-stimulus condition and inversely related to stimulus
attraction (Fig. 1d).

To quantify these trends, we calculated the average bulk $c$-$fos$ inten-
sity per cluster in each animal (Extended Data Figs. 1e and 2c). Statistical
analysis of the $c$-$fos$ signal revealed significant modulation of activity in
21 clusters by at least one stimulus relative to the no-stimulus condition

(Fig. 1e). Hierarchical clustering of all areas separated five major groups
of clusters that were qualitatively (1) not modulated relative to the
absence of stimulus, (2) weakly suppressed by virtual conspecifics,
(3) suppressed by most stimuli, (4) activated more by real conspecifics
than virtual ones and (5) activated by most stimuli (Fig. 1e).

Together, this unbiased global activity map identified brain net-
works whose activity is modulated by real and virtual conspecifics with
putative roles in social affiliation. These include posterior preoptic
and rostral hypothalamic areas that are likely to be homologous to

paraventricular and anterior hypothalamic nuclei commonly assigned to the conserved social behaviour network[12–14] (Hrl, mHr, PPp). Clusters in groups 3 and 4, which were modulated exclusively by real conspecifics (Hc3, Hi2, PPp, Hc2), might reflect neuronal responses beyond those necessary for acute social affiliation. They may contribute to perception of threat or homeostatic stress mechanisms, potentially through visual cues that are not present in bout-like dot stimuli or additional sensory modalities such as olfaction and mechanosensation[16,21,25,26]. Thus, the set of clusters activated by virtual conspecifics highlights a core network underlying the visuomotor transformation associated with shoaling, beginning with the recognition of conspecifics.

## Thalamic neurons encode biological motion

Our *c-fos* labelling method highlights putative visual input pathways for social affiliation. Biological motion probably enters the brain through the TeO and DT[27], therefore providing an opportunity to investigate sensory detection of this social cue. To understand stimulus selectivity of individual neurons in these visual areas, we turned to volumetric two-photon calcium imaging of juvenile brain activity in response to presentation of virtual conspecifics.

Fish that expressed nuclear-localized GCaMP6s in almost all neurons (carrying the transgene *elavl3:H2B-GCaMP6s*) were immobilized on the stage of a microscope equipped with a custom-built remote focusing set-up for rapid image acquisition (Extended Data Fig. 3). We imaged simultaneously in 6 imaging planes at 5 Hz, extending $600 \times 600 \times 200$ um ($x,y,z$) (Fig. 2a). This volume included the retinorecipient brain areas highlighted by our *c-fos* analysis, DT and TeO, as well as pretectum, nucleus isthmi, VT and habenulae (Fig. 2b and Extended Data Fig. 4). Analysis of 28,306 registered neurons across 11 animals revealed that responses to virtual conspecifics were most prominent in the TeO (51% of active neurons), followed by pretectum (12%), DT (10%) and nucleus isthmi (10%) (Extended Data Fig. 4), qualitatively matching the *c-fos* mapping results (Fig. 2c).

To identify neurons that encode biological motion, we computed a bout preference index (BPI) as the normalized difference in the response to behaviourally attractive bout-like motion versus unattractive continuous motion (Fig. 2d,e). The majority of neurons did not differentiate between bout frequencies (mean BPI $0.01 \pm 0.07$). However, $13 \pm 5\%$ of all neurons scored BPI $> 0.5$, corresponding to a threefold increase in $\Delta F/F$ for bout-like motion compared with continuous motion in these neurons. We next focused our attention on these putative bout preference neurons (BPNs). In a subset of animals, we determined that the BPN population was largely unresponsive to looming and moving grating control stimuli (Extended Data Fig. 5a). Most BPNs were located in the TeO (36%) and DT (18%) (Extended Data Fig. 4). Gaussian kernel density estimation (KDE) yielded the DT as the anatomical area of highest BPN density (Fig. 2f). Within the DT, BPNs were concentrated in a posterior cluster, overlapping with DT *c-fos* activity (Fig. 1c). By contrast, tectal BPNs were distributed broadly along the anteroposterior and dorsoventral axes with lower relative frequency (Extended Data Fig. 4c). The anatomical overlap of *c-fos* and GCaMP signals in the TeO and DT suggests that virtual conspecifics in the open-loop configuration activate key circuits for social recognition even in immobilized animals.

If activity of DT-BPNs drives shoaling behaviour, their tuning should match specific parameters of biological motion. DT-BPNs had a response peak at a stimulus bout frequency of $1.2 \pm 1.6$ Hz, closely matching the juvenile's typical swim bout frequency of around 1.25 Hz, which also most effectively elicits shoaling[2] (Fig. 2g). To examine whether DT-BPNs encode acceleration or average speed of virtual conspecifics, we collected a separate dataset and systematically varied each parameter independently (Fig. 2h). At continuous motion, DT-BPNs were barely modulated by stimuli moving at 2 to 150 mm s$^{-1}$. At 1.5 Hz, DT-BPNs yielded maximal responses at $7.2 \pm 1.7$ mm s$^{-1}$ (Fig. 2h), similar to a juvenile's typical swim speed at around 5 mm s$^{-1}$ and, again,

matching the behavioural tuning[2]. Morphing acceleration from continuous to bout-like along Gaussian speed profiles at fixed average speed of 5 mm s$^{-1}$ and 1.5 Hz bout frequency modulated DT-BPN responses as a function of acceleration with a maximum at the highest possible acceleration of 12 m s$^{-2}$ (projector limit). Taken together, DT-BPNs detect biological motion through periodic acceleration at fish-like speed and bout frequency, and are therefore tailored for the detection of juvenile zebrafish during shoaling.

To relate DT-BPN responses to naturalistic visual percepts, we tested another set of animals with 'dot shoaling' stimuli on trajectories that recapitulate positions of conspecifics relative to a real focal animal during shoaling (Extended Data Fig. 5b). DT-BPNs were strongly and persistently activated by such stimuli (Extended Data Fig. 5d). Self-motion during shoaling also generates global motion with temporal dynamics similar to the fish-like cues. To examine whether global motion activates DT-BPNs, we rotated whole-field stimuli with matched bout-like motion and spatial frequency (Extended Data Fig. 5c). Global motion strongly activated pretectal neurons[27] but not DT-BPNs (Fig. 2i), suggesting that the latter encode fish-like biological motion and not self-motion-induced visual signals.

Zebrafish shoaling with real or virtual conspecifics emerges at around two weeks of age[2,6,28], whereas younger fish show mainly interanimal repulsion[2,29]. We therefore hypothesized that functional maturation of BPNs coincides with this transition. Contrary to this prediction, BPNs already existed in larvae, but with lower fractions compared with the juveniles ($10 \pm 3\%$ of all recorded neurons). Furthermore, BPNs were similarly distributed in the brain, with the KDE centre in the DT (Fig. 2j). Registration of the larval data to the Max Planck Zebrafish Brain Atlas[30,31] (https://mapzebrain.org) confirmed localization of the DT-BPN cluster to the *vglut2a*-positive DT area, ventrally touching the *gad1b*-positive VT[32]. The larval DT-BPN cluster was molecularly defined by the expression of *cort*, as seen in juveniles, and *pth2* (Extended Data Fig. 5e), a gene of which thalamic expression tracks the density of conspecifics in zebrafish through mechanosensory signals[33]. Overlap with *pth2* raises the possibility of multimodal integration of conspecific signals in DT. Furthermore, the mean frequency tuning curve across all larval DT-BPNs and mean tuning peak were similar to juveniles (Extended Data Fig. 5f). Finally, we determined that DT-BPNs also developed in socially isolated larvae, demonstrating that tuning to conspecific motion in DT is largely independent of social experience (Extended Data Fig. 5g). Thus, functional BPN maturation precedes shoaling, and the developmental transition is either gradual in nature or requires a change in additional circuit nodes. The presence of BPNs in pre-juvenile stages provides an opportunity to investigate the circuit with the experimental tools and resources available in larvae.

## EM reconstruction of the biological motion circuit

Across vertebrates, the thalamus acts as a gateway for state-dependent sensory information[32,34]. We hypothesized that DT-BPNs could serve that role for social cues, connecting visual brain areas and the conserved social behaviour network[12,27]. To reveal the anatomy of the DT-BPN circuit, we analysed an electron microscopy (EM) whole-brain dataset of a larval zebrafish at 5 days post-fertilization (d.p.f.), acquired at synaptic resolution[35]. We registered the larval DT onto the EM volume to identify the cell body location of putative BPNs (pBPN) in the DT (Extended Data Fig. 6a–c). We randomly selected and completely traced 34 cells in this region (Fig. 3a,b). All of these cells extended their primary neurite ventrolaterally and showed both dendritic and axonal arborizations in a thalamic neuropil region, posterior to retinal arborization field AF4 (Fig. 3b,c). In this region, we randomly selected presynaptic contact sites on putative BPN dendrites, supported by an automated synapse segmentation[35] (Methods), and identified their partner neurons (Fig. 3c). A total of 26.7% of input synapses were provided by other DT neurons (Extended Data Fig. 6d) and, similarly, 26.7% arrived from

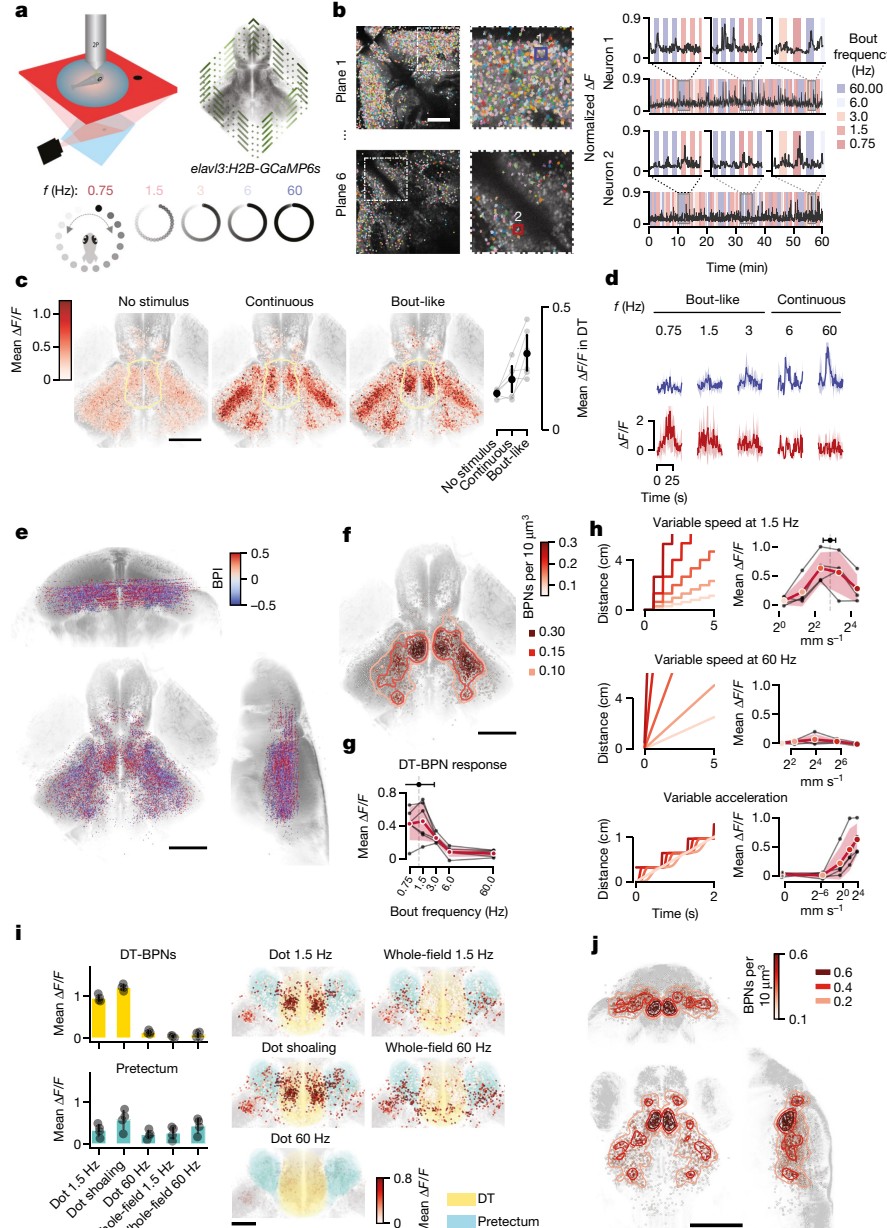

**Fig. 2 | Dorsal thalamus neurons are activated by fish-like motion.**
**a**, Schematic of the experimental set-up. **b**, Example imaging planes in the TeO and DT with all segmented neuronal ROIs, representative for *n* = 11 animals (left). Right, representative normalized Δ*F*/*F* traces of one tectal and one thalamic neuron. **c**, Horizontal view of all responsive neurons (*n* = 28,306 total, 2,573 ± 1,175 per fish) from 11 fish (18–22 d.p.f.) aligned to a juvenile reference brain (left). Colour indicates mean baseline Δ*F*/*F* (no stimulus), and responses to continuous and bout-like motion. The yellow line indicates the DT. Right, mean responses of all DT neurons per fish (*n* = 258 ± 198, 2,837 total) from *n* = 6 animals with a number of recorded DT-BPNs of >30. Data are mean ± 1 s.d. **d**, Mean Δ*F*/*F* responses of example neurons from **b** to all stimulus frequencies. **e**, The distribution of all responsive neurons from *n* = 11 fish in the reference brain. The colour map shows the BPI. Opacity scales with absolute BPI (0–0.5). **f**, The distribution of BPNs (312 ± 143 neurons per fish, 3,437 total). Colour

reflects a Gaussian KDE; contours delineate densities of 0.1, 0.15 and 0.3 BPNs per 1,000 μm³. *n* = 11 fish. **g**, DT-BPN tuning to stimulus frequency. The mean peak across neurons was 1.2 Hz ± 1.6 Hz. *n* = 563 neurons. The black lines represent the mean values of individual animals. Data are from a subset of animals in **e** with a number of recorded DT-BPNs of >30. *n* = 6 animals. **h**, DT-BPN tuning to average speed at 1.5 Hz or 60 Hz and acceleration. The cartoons show stimulus displacement over time. Data are mean ± 1 s.d. of all of the neurons shown above. *n* = 291 neurons. The black lines indicate individual animals. *n* = 4 fish, 73 ± 10 neurons per fish. **i**, DT-BPN and PreT responses to local dot motion and whole-field motion and their anatomical distribution. Circles (left) show the mean of individual animals. *n* = 4 fish, 77 ± 15 (DT-BPNs), 114 ± 48 (PreT) neurons per fish. Data are mean ± 1 s.d. **j**, The distribution of BPNs in 7 d.p.f. larvae (*n* = 4 fish, 230 ± 87 neurons per fish) as in **f**. Scale bars, 100 μm (**b** and **i**) and 200 μm (**c**, **e**, **f** and **j**).

tectal periventricular projection neurons (PVPNs; Fig. 3d). We also identified synaptic input from the VT (16.7%), ipsi- and contralateral nucleus isthmi (10%; Extended Data Fig. 6e), superior ventral medulla oblongata (6.7%), torus semicircularis (6.7%), hypothalamus (3.3%) and cerebellum (3.3%). Overall, the majority of input synapses were established by ipsilateral neurons (76.7%). Identified PVPNs (*n* = 13)

send their axons ventrally through the postoptic commissure, and make ipsi- or contralateral connections to putative BPNs within the thalamic neuropil region. A single PVPN can be presynaptic to several putative BPNs (Fig. 3e), and a single BPN can receive input from several PVPNs, both ipsi- and contralaterally (Extended Data Fig. 6f). Next, we quantified presynaptic partners of DT-projecting PVPNs: 55% of all input

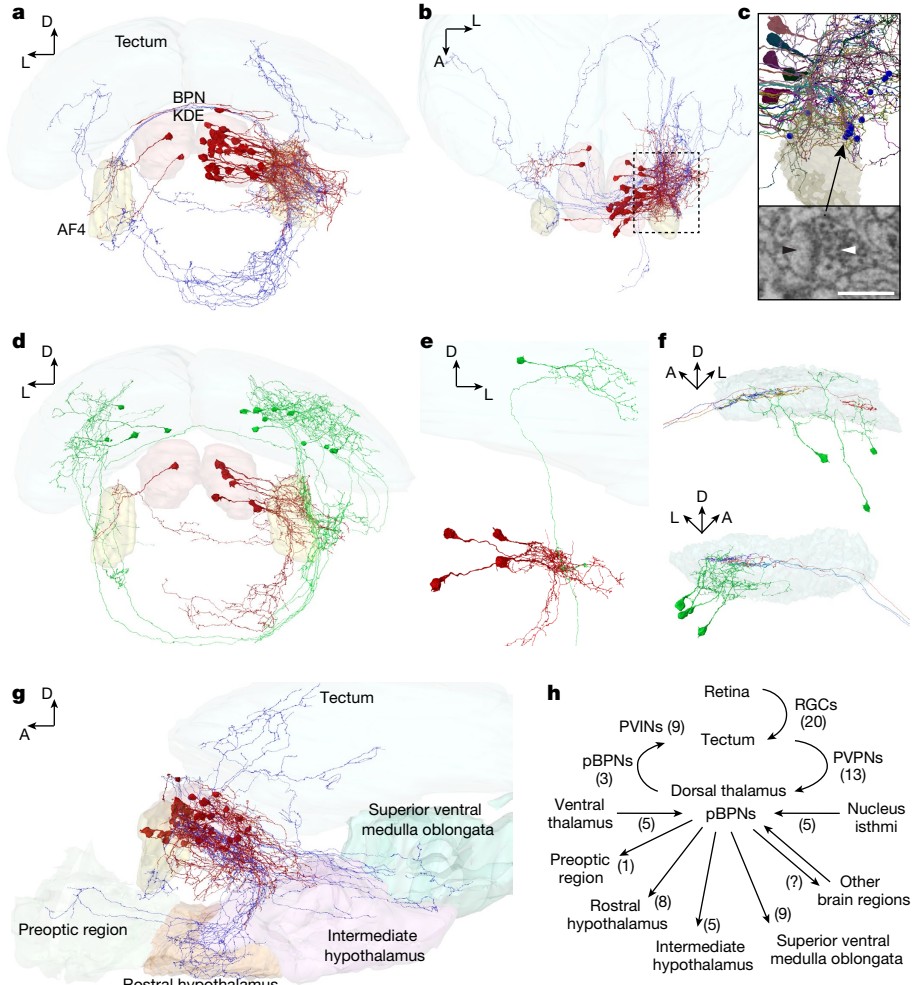

**Fig. 3 | Connectivity of the larval thalamic bout-preference region.**
**a**, Frontal view of an EM reconstruction of neurons in the bout-preference region (BPN KDE, red) of the DT. Axons are shown in blue. **b**, Top view of the neurons shown in **a**. **c**, Magnified view of the thalamic arborization field, outlined in **b**. Synapses with identified presynaptic partners are shown as blue spheres. One representative synapse of a tectal PVPN axon (white arrow) onto a putative BPN's dendrite (pBPN, black arrow) is indicated below (randomly chosen). **d**, Frontal view of tectal PVPNs (green) and their postsynaptic pBPN partners (red). **e**, Example of a single PVPN (green), which makes ipsilateral synaptic contacts to at least four identified pBPNs (red). **f**, Side view of the left (top) and the right (bottom) tectal SFGS layers, showing the PVPNs (green) and their presynaptic retinal ganglion cell axons (different colours). PVPN axons are not shown for clarity. **g**, Side view of the pBPNs (red, axons in blue) and their axonal target regions (Supplementary Video 2). AF4 (yellow) is shown as a reference. **h**, Circuit diagram. Identified cell types are indicated next to arrows with cell numbers in parentheses. For **c**, scale bar, 0.5 μm.

synapses arrived from a specific class of retinal ganglion cells, which exclusively arborized in the SFGS3/4 layer of the tectum[36] (Fig. 3); 35% were from tectal periventricular interneurons and, interestingly, 10% of input synapses arrived from other PVPNs, which also projected to the DT and contacted pBPNs.

We further investigated the downstream target regions of putative DT-BPNs. Of the 34 traced DT neurons, 24 had long projection axons and made synaptic contacts in other brain areas, whereas 10 neurons had local (*n* = 3) or premature (*n* = 7) axonal projections. Our analysis revealed the superior ventral medulla oblongata (*n* = 9 cells), rostral hypothalamus (*n* = 8), intermediate hypothalamus (*n* = 5), contralateral thalamus (*n* = 3), preoptic region (*n* = 1) and the tectum (*n* = 3) as axonal targets (Fig. 3g,h). Putative BPNs that projected to the tectum targeted the SFGS layer, where they contacted tectal periventricular interneurons (Extended Data Fig. 6g).

To complement the EM tracings, we next analysed the morphology of traced neurons residing in the BPN-KDE of the light microscopy map-zebrain atlas[30]. We identified 13 putative BPNs that all extended their primary neurites ventrolaterally into a neuropil area posterior of AF4, consistent with the EM data. Of these neurons, 12 projected into other

brain areas, including the tectum (*n* = 1), preoptic area (*n* = 1), rostral hypothalamus (*n* = 1), intermediate hypothalamus (*n* = 3), superior ventral medulla oblongata (*n* = 5) and inferior ventral medulla oblongata (*n* = 2) (Extended Data Fig. 6h,i).

These findings suggest a pathway for the detection of biological motion: retinal information reaches DT-BPNs through tectal PVPNs and is subsequently transmitted to brain areas that are proposed to regulate social behaviour, including the preoptic region, and clusters in the rostral, intermediate and caudal hypothalamus, which showed *c-fos* signal during shoaling behaviour (Fig. 1f). Gradual maturation of DT projections and/or addition of synapses, such as those connecting the ventral forebrain at around 14 d.p.f., may then underlie the emergence of shoaling at the juvenile stage[37].

## Social attraction requires the biological motion circuit

As TeO and DT-BPNs are activated by fish-like motion, we hypothesized that this pathway is necessary for shoaling. To test this hypothesis, we ablated TeO and DT in juvenile animals and analysed effects on free-swimming interactions with virtual conspecifics.

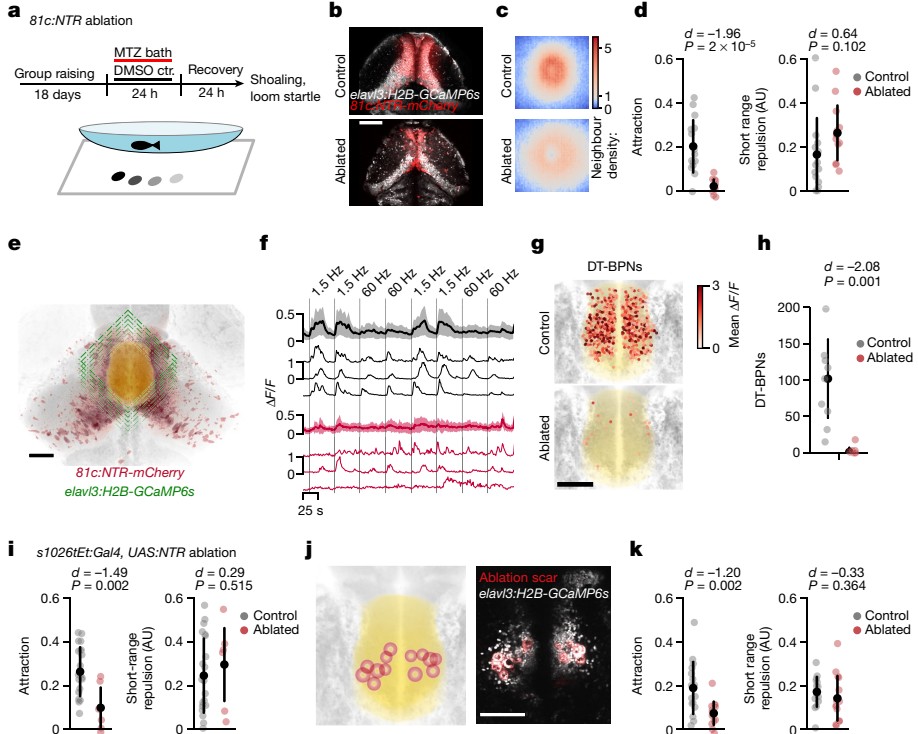

**Fig. 4 | The TeO–DT circuit is necessary for social attraction. a**, Schematic of the shoaling test after chemogenetic ablation. ctr., control. **b**, Two-photon image of 21 d.p.f. *SAGFF(lf)81c:Gal4, UAS:NTR-mCherry* and *elavl3:H2B-GCaMP6s* animals 24 h after ablation versus control treatment. Representative of three similar fish. **c,d**, Reduced neighbour density (**c**) and attraction (**d**) in *81c:NTR*-ablated animals. Short-range repulsion is intact. *n* = 13 (ablated) and *n* = 15 (control) animals. The neighbour maps in **c** show the probability of finding the stimulus in space with the animal at the centre of the map, heading up. Each map is 60 mm × 60 mm. AU, arbitrary units. **e**, Schematic of volumetric two-photon imaging in the DT after *81c:NTR* ablation. **f**, Mean and example Δ*F* traces of all DT-BPNs for ablated and control animals show that *81c:NTR* ablation strongly reduces responses to bout-like motion. The vertical lines mark the start of stimulus presentation. The shaded areas denote 1 s.d. around the mean. *n* = 25 (ablated) and *n* = 849 (control). **g,h**, Fewer DT-BPNs in

*81c:NTR (-*ablated) animals. **g**, The anatomical location of all DT-BPNs across animals coloured by mean Δ*F/F* to bout-like motion. *n* = 25 (ablated) and *n* = 849 (control). The yellow area shows the DT. **h**, Quantification of DT-BPNs per animal. *n* = 7 (ablated) and 9 (control) animals. Cohen's *d* effect size is shown. The *P* value was calculated using the two-sided Mann–Whitney *U*-test. **i**, Attraction is strongly reduced in *s1026tEt:NTR*-ablated animals. Short-range repulsion is intact. *n* = 7 (ablated), *n* = 21 (control) animals. **j**, Bilateral two-photon laser ablation of neurons in the DT-BPN region in juvenile zebrafish. **k**, Reduced attraction after DT laser ablation. *n* = 7 (ablated) and *n* = 9 (embedding control) animals. Short-range repulsion is intact. Data in **d**, **h**, **i** and **k** represent individual animals and mean ± 1 s.d. Cohen's *d* effect size is shown. The *P* values were calculated using two-tailed Student's *t*-tests with no correction. For **b**, **e**, **g** and **j**, scale bars, 100 μm.

We genetically targeted tectal cells for chemogenetic ablation using the *SAGFF(lf)81c* enhancer trap line to drive expression of nitroreductase. *SAGFF(lf)81c* is strongly expressed in tectal neurons and weakly expressed in parts of the pretectum, habenula, anterior DT and anterior VT (Fig. 4b and Extended Data Fig. 8). Although ablated animals appeared healthy and had slightly faster swim kinetics, they showed a severe loss of attraction towards virtual conspecifics (*P* < 0.001; Fig. 4c,d and Extended Data Fig. 7). To investigate the spatial scale of this behavioural defect, we computed neighbour density maps that represent relative spacing with virtual conspecifics. In the controls, neighbour maps revealed a central zone of short-range (5–15 mm) repulsion, surrounded by a ring of long-distance attraction (10–30 mm). In ablated animals, this balance was shifted. The ring of attraction was strongly reduced, whereas the zone of repulsion was intact (Extended Data Fig. 7a,b). Moreover, looming-induced startle responses were at the control level in ablated animals (Extended Data Fig. 7c). Finally, we confirmed that *SAGFF(lf)81c* ablation disrupted shoaling with a real conspecific (*P* < 0.001; Extended Data Fig. 7d).

To understand the neural correlates of these effects, we recorded neuronal responses to continuous and bout-like motion in DT after *SAGFF(lf)81c* ablation. We found that ablation reduces the number of DT-BPNs by more than 95% (*P* = 0.0014, 4 ± 6 versus 94 ± 4 cells per animal) (Fig. 4f,g). We observed similar trends for dot motion at 1.5 and 60 Hz

(continuous), at which tectal ablation significantly reduced the number of top-scoring neurons (*P* < 0.01). Responses to looming and translational grating motion in the surrounding pretectum were also reduced in ablated 21 d.p.f. juveniles but not in ablated 7 d.p.f. larvae (Extended Data Fig. 8). These results are consistent with our inferred wiring diagram, which places tectal PVPNs upstream of DT-BPNs as their main sensory driver.

Next, we tested the necessity of DT for shoaling by chemogenetic ablation using the *s1026tEt* enhancer trap line to drive expression of neuronally restricted nitroreductase. The *s1026tEt* Gal4 insertion drives strong and selective expression of a UAS-linked nitroreductase transgene in DT of juvenile zebrafish (Extended Data Fig. 8a). We found that ablation of *s1026tEt* cells in juveniles caused a selective loss of attraction (*P* = 0.002) and a modest increase in swim bout frequency, whereas short-range repulsion and visual escapes remained intact (Fig. 4i and Extended Data Fig. 7e). Finally, we tested the necessity of DT-BPNs directly by laser-ablation of the DT-BPN area in 21 d.p.f. animals. After ablation, we found a selective loss of attraction, whereas short-range repulsion, visual escapes and overall swim kinetics remained intact (*P* = 0.002; Fig. 4j,k and Extended Data Fig. 7e). Together, these results suggest that *SAGFF(lf)81c* neurons and DT-BPNs are essential elements of a pathway that mediates the affiliative aspect of shoaling, but are dispensable for collision avoidance during shoaling and visual escape from a looming threat.

## Discussion

Affiliation with conspecifics is a core building block of social behaviours that offer benefits for collectives, such as efficient food detection or evasion of predators[3,8,38]. As a consequence, animals need to robustly recognize neighbours in cluttered environments during highly dynamic interactions. These interactions rely on the balance of attraction and repulsion into an appropriate distance[1,4,5]. Although empirical models of collective behaviour have postulated distinct individual-level behavioural rules[5,7,8], the neuronal implementation of such coordination has remained unclear, largely because mutual interactions mask causal relationships between conspecific signals and receiver responses. Our results in shoaling zebrafish now highlight fish-like motion[2,11] as a salient trigger signal of an attraction pathway that converges on a multimodal[33], socially activated DT cluster and feeds into hypothalamic areas that are probably homologous with nodes of the proposed social behaviour network[12–14,21]. Neuronal activity in this circuit therefore represents an inherently kinetic metric of neighbouring animals, in contrast to current shoaling models that emphasize positional information[1,4,6,20]. By contrast, short-range repulsion engages a separate circuit, probably overlapping with the collision-avoidance pathway[39–41]. These results add support for the emerging importance of visual motion cues in social recognition[10,42–44]. The correspondence of sensory activation in freely shoaling versus immobilized animals with virtual conspecifics suggests that this approach can also reveal the role of neural circuit nodes in the downstream network during shoaling for an understanding how collective dynamics emerge from neuronal computations in individuals.

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

# Methods

## Animal care and transgenic zebrafish

Adult, juvenile and larval zebrafish (*Danio rerio*) were housed and handled according to standard procedures. All animal experiments were performed under the regulations of the Max Planck Society and the regional government of Upper Bavaria (Regierung von Oberbayern), approved protocols: ROB-55.2Vet-2532.Vet 03-15-16, ROB-55.2Vet-2532.Vet 02-16-31, and ROB55.2Vet-2532.Vet 02-16-122. Experimental animals were outcrosses to TL or TLN (nacre) unless otherwise noted. The following transgenic lines were used: *Tg(elavl3:H2B-GCaMP6s)jf5*[45], *SAGFF(lf)81c* (TeO Gal4 line)[46], Tg(*UAS-E1B:NTR-mCherry)c264*[47], *Et(fos:Gal4-VP16)s1026t* (DT Gal4 line)[48], *Tg(UAS:BGi-epNTR-TagRFPT-utr.zb3)mpn420* (this study).

Larvae were raised in Danieau solution under a 14–10 h light–dark cycle at 28.5 °C until 6 d.p.f. For experiments in juveniles, animals were then raised under standard facility conditions at 28.5 °C in groups of 20–25 individuals. The fish were fed by feeding robots once a day with artemia and 2–3 times a day with dry food.

## Shoaling assay and behaviour quantification

Shoaling with real and virtual conspecifics was assayed as previously described[2]. In brief, 15 or 35 individual animals were transferred individually into shallow watch glass dishes of 10 cm or 7 cm diameter, respectively, separated by a grid of visual barriers and resting on a projection screen. Custom-written Bonsai[49] workflows were used to project stimuli to each animal and to track animal location at 30 fps. Stimuli were black dots on a white background moving along a predefined, synthetic-trefoil shaped trajectory at an average speed of 5 mm s$^{-1}$. For continuous motion, the stimulus position was updated 30 times per second. For bout-like motion, the stimulus position was updated once every 666 ms. Dot diameter was 2 mm unless noted otherwise, and 0 mm in the no-stimulus condition.

To assay shoaling of pairs of real conspecifics, we introduced a second animal in the same dish and did not show any projected stimuli. Fast-Track[50] was used for post hoc tracking of real pair shoaling.

Attraction and neighbourhood maps were quantified as previously described[2] using custom-written Python software. We calculate the 'real' average interanimal distance or animal dot distance for each animal in 5 min chunks (IADr). Next, we generate 10 time-shifted trajectories and recalculate the shifted average inter animal or animal dot distance (IADs) for each time shift. Mean IADs for all time shifts are used to compute attraction as (IADs – IADr)/IADs.

For neighbourhood maps, neighbour position time series were transformed into the focal animal's reference frame to compute a binned 2D histogram.

Repulsion was quantified as the reduction in attraction at the centre of each animal's neighbour density map. Neighbour density maps were gaussian-filtered (sigma = 3 mm) before obtaining 24 radial line scans (width of 5 mm) starting from the centre of the map. Repulsion was the area above the average line scan, at radii less than the radius at which maximum neighbour density occurred (Extended Data Fig. 7a), divided by the full length of the scan (29 mm).

Looming stimuli were presented in the virtual shoaling setup[41]. Looming discs appeared once every minute at a defined offset of 5 mm to the left or the right from the current centre of mass of each animal. Looming discs expanded within 500 ms to the indicated final size and followed the animal. To compute an escape fraction, we defined an escape response as a trial in which the animal moved more than twice as far in a time window of 1 s immediately following the loom compared to the 1.3 s before. Bout duration was computed using peak detection on the velocity time series of each animal.

## c-fos activity mapping

**Shoaling assay for c-fos.** For *c-fos* labelling, we used *nacre;elavl3:H2B-GCaMP6s* fish at 21 d.p.f. Thirty five fish were transferred into individual dishes and left without stimulation in the presence of white projector illumination from below for acclimatization and to establish a low, non-social *c-fos* baseline. Each animal was assigned randomly to one of the four stimulus groups. After 2 h, continuous or bout-like motion were shown to groups 1 and 2, respectively, whereas groups 3 and 4 continued to see no stimulus. After 45 min, groups 1, 2 and 3 were quickly euthanized and fixed. Four animals of group 4 were then transferred into the dishes of four other animals of this group for shoaling. After 45 min, these eight animals were euthanized and fixed as well.

**HCR staining and imaging.** Animals were euthanized and fixed on 4% ice cold paraformaldehyde (PFA). The PFA was washed out after 24 h with 1× PBS and the samples were gradually dehydrated and permeabilized with methanol and stored in −20 °C for several days until the HCR in situ labelling was performed. All of the HCR reagents were purchased from Molecular Instruments and the staining was performed according to the manufacturer's protocol for whole-mount zebrafish larvae. In brief, the samples were separated into 2 juvenile fish per single 1.5 ml Eppendorf tube. Rehydration steps were performed by washing for 5 min each in 75% methanol/PBST (1× PBS + 0.1% Tween-20), 50% methanol/PBST, 25% methanol/PBST and finally five times with 100% PBST. The samples were permeabilized with 30 µg ml$^{-1}$ proteinase K for 45 min at room temperature, followed by postfix with 4% PFA for 20 min at room temperature and 5 washes in PBST for 5 min each. The samples were prehybridized in a 500 µl probe hybridization buffer (Molecular Instruments) for 30 min at 37 °C. Hybridization was performed by adding 2 pmol of each probe set to the hybridization buffer and incubating for 16 h at 37 °C. Probe sets for *c-fos* (B5 initiator), *cort* (B3 initiator) and *elavl3* (B2 initiator) were purchased from and designed by Molecular Instruments. To remove the excess probes, the samples were then washed 4 times for 15 min each with a wash buffer (Molecular Instruments) at 37 °C, followed by 2 washes of 5 min each with 5× SSCT (5× SSC + 0.1% Tween-20) at room temperature. Pre-amplification was performed by incubating the samples for 30 min in an amplification buffer (Molecular Instruments) at room temperature. The fluorescently labelled hairpins (B2-488, B3-647, B5-546) were prepared by snap cooling: heating at 95 °C for 90 s and then cooling to room temperature for 30 min. Hairpin solution was prepared by adding 10 µl of the snapped-cooled hairpins (3 µM stock concentration) to a 500 µl amplification buffer. The pre-amplification buffer was removed, and the samples were incubated in the hairpin solution for 16 h at room temperature. The excess hairpins were washed three times with 5× SSCT for 20 min each wash, and the samples were stored in 5× SSCT in the dark at 4 °C until imaging.

For dorsal imaging, the samples were embedded in 2.5% low melting agarose in 1× PBS. Imaging was performed with a Leica SP8 confocal microscope equipped with a ×20 water-immersion objective. *z*-Stacks, comprising four tiles, covering of the entire brain were taken (final stitched image size: 1,950 px × 1,950 px, 1,406 µm × 1,406 µm, 3 µm in *z*). All 32 samples were imaged with the exact same laser power, gain, zoom, averaging and speed to faithfully quantify and compare the fluorescent signal between the samples. For ventral imaging, the samples were removed from the agarose and dissected to remove the jaw and the gills. After the dissections, the samples were embedded upside down and imaged in the same manner. Four brains were lost during ventral imaging and were therefore excluded entirely from the subsequent analysis.

**Image registration.** Image registration was performed using Advanced Normalization Tools (ANTs)[51] running on the MPCDF Draco/Raven Garching computing cluster. Before registration, stacks were batch-processed in ImageJ. Each stack was downsampled to 512 px width at the original aspect ratio using bilinear interpolation, split into individual channels and saved as .nrrd files. For ventral stacks, artefacts of the dissection such as left-over autofluorescent muscle fibres

and skin were masked before registration. Initial attempts to register the *elavl3* HCR channel of dorsal or ventral HCR confocal stacks to a live-imaged two-photon reference of *elavl3-H2B-GCaMP6s* expression were not successful, probably due to deformations resulting from the HCR protocol and diverse qualitative differences in image features between the imaging modalities. Instead, separate dorsal and ventral HCR registration templates were generated from scratch by running antsMultivariateTemplateConstruction2.sh on three manually selected stacks. Next, all dorsal and ventral stacks were registered to their respective templates using antsRegistration. Finally, the ventral template was registered to the dorsal template using affine + b-spline transformations via antsLandmarkBasedTransformInitializer with the help of 25 manually curated landmarks in each stack before applying standard antsRegistration. The resulting ventral-to-dorsal transform was then applied to re-register all ventral stacks into one common (dorsal) reference frame.

**c-fos signal intensity quantification.** Image analysis was performed using custom scripts in Python. Registered dorsal and ventral stacks were merged as the arithmetic mean intensity for each animal. To normalize to a drop in signal intensity with tissue depth, the *c-fos* signal was divided voxel-wise by the *elavl3* HCR signal. For visualizations of imaging planes, the *elavl3* signal used in normalization was filtered by a 3D gaussian (filter width: 55 µm, 55 µm, 15 µm $x,y,z$). For area-wise *c-fos* quantification, unfiltered *elavl3* signal was used in normalization. To identify activity clusters, merged stacks from all animals per condition were generated by finding the maximum intensity at each voxel across animals. A combined RGB hyperstack was generated that showed *c-fos* signal for each condition, *cort* HCR and *elavl3* HCR for reference in different colours for visual inspection. Activity clusters were manually drawn as 3D masks on the hyperstack using the ImageJ segmentation editor on orthogonal overlay views. Masks were drawn with the intent to outline prominent, distinct clusters of *c-fos* signal, irrespective of their modulation by social condition. The full hyperstack, including cluster masks is available. Brain areas housing the activity clusters were identified by comparison of the *elavl3* reference to the mapzebrain atlas[30] and additional resources[13,52,53].

Individual *cort*- and *c-fos*-positive cells in DT were counted manually using the ImageJ cell counter plugin. For statistical analysis across activity clusters and conditions, bulk normalized *c-fos* signal was computed as the average intensity of all voxels belonging to a given cluster. Effect size was determined in each cluster for each condition versus the no-stimulus condition by pairwise computation of Cohen's $d$ defined as the difference of the means divided by the pooled standard deviation. To determine significant activity modulation compared to the no-stimulus condition, we performed repeated two-tailed $t$-tests and corrected for multiple comparisons in each family of tests (each activity cluster) using the Bonferroni correction. Hierarchical clustering of the activity clusters was performed on the effect sizes using the seaborn method clustermap with the default parameters for average Euclidean clustering.

## Functional two-photon calcium imaging

Two-photon functional calcium imaging was performed on 6–8 d.p.f. larvae and 17–22 d.p.f. juvenile *elavl3:H2B-GCaMP6s* transgenic fish without paralysis or anaesthesia. The 6–8 d.p.f. larvae were embedded in 2% agarose with the tail freed as previously described[39]. Juveniles (17–22 d.p.f.) were embedded in 3% agarose. As juvenile zebrafish are prone to hypoxia in this preparation, several precautions were taken. A drop of low-melting agarose was placed onto a petri dish and allowed to cool before a fish was introduced and oriented with a pipette tip. Once solidified, agarose was removed from the mouth, gills and tail using scalpels to restore active and passive breathing (Extended Data Fig. 3). Additional oxygen was supplied by continuously perfusing the dish. The perfusion medium consisted of fish water freshly oxygenated

to saturation at the start of the experiment and diluted 1:1 with demineralized water to support ionoregulation and buffered with 1.2 mM $NaH_2PO_4$ and 23 mM $NaHCO_3$[54]. To monitor health, we checked heartbeat and breathing movements of gills and mouth before and after an experiment. Only fish that were breathing and moving after the end of the experiment were included in the analysis. The embedded fish were mounted onto the stage of a modified two-photon moveable objective microscope (MOM, Sutter Instrument, with resonant-galvo scanhead) with a ×20 objective (Olympus XLUMPLFLN, NA 1.0) and imaged for at least 60 min. Typically, fish resumed swimming immediately after release from embedding. Only fish that did not drift up or down in their preparation were used for analysis. Fish in which no tectal responses could be observed were eliminated from the analysis. Fast volumetric imaging of the tectum and/or thalamus was performed using a custom-built remote focusing arm added before the microscope (Extended Data Fig. 3b). The remote focusing path was constructed using the 30 mm and 60 mm Thorlabs cage system, and consists of the following parts (in order of forwards traversal): a half-wave plate (Thorlabs, AHWP05M-980), a polarizing beam splitter (Thorlabs, PBS102), two lenses (Thorlabs, AC254-100-B-ML and AC508-200-B-ML), a quarter-wave plate (Thorlabs, AQWP10M-980), remote objective (Nikon, CFI ×16 0.8 NA), and a gold mirror (Thorlabs, PF05-03-M01) mounted onto a custom piezo (PPS-D08300-001 nanoFaktur, 300 µm closed loop range, with a nPoint LC.402 controller). The piezo was mounted on a *xyz* translation stage with tip-tilt control. Changing the mirror position is rapid (for the step sizes used for imaging, 1–2 ms) and results in a change of focus of the excitation beam exiting the main objective. Refocusing through the remote arm enabled rapid sequential imaging of 6 planes with a nonlinear step size ranging from 6–24 µm at 5 volumes per second. Remote focusing was not used for the high-resolution single-plane imaging in Fig. 2h,i. The plane size ranged from 370 µm × 370 µm for larvae to 1,075 µm × 1,075 µm for juveniles. Laser power ranged from 12.3 mW to 15.4 mW. The spatial sampling (0.7–2.1 µm px⁻¹) and optical resolution enabled discrimination of single cells with cell body diameters typically in the range of 5 µm to 8 µm.

**z-Stack acquisition and image registration.** For each functionally imaged fish, a z-stack of the entire brain was taken (512 × 512 or 1,024 × 1,024 pixels, 2 µm in z, 835–920 nm laser wavelength, plane averaging 50–100×) with the two-photon microscope. Larval data were registered to the mapzebrain atlas[30] using the *elavl3:H2B-GCaMP6s* reference. For juvenile data, a standard brain was generated from three high-quality z-stacks (150× frame averaging) as described in the '*c-fos* activity mapping' section and each juvenile brain was registered to it. The generation of a standard brain and the parameters used for ANTs registration have been described in detail previously[30].

To align functional regions of interest (ROIs) from 2P data to a common reference frame, a two-step strategy was used. First, average frames of all imaging planes were registered to individual z-stacks using template matching. Converted ROI locations in z-stack coordinates were then transformed to the larval and juvenile common reference frames by running the ANTs command antsApplyTransformsToPoints with the matrices from the z-stack registrations.

**Visual stimuli.** Visual stimuli were designed using PsychoPy and projected by an LED projector (Texas Instruments, DLP Lightcrafter 4500, with 561 nm long-pass filter) on Rosco tough rolux 3000 diffusive paper placed into a petri dish filled with fish water.

**Frequency tuning.** A black dot moving on a circular trajectory (radius, 18 mm) with the fish head in the centre was shown starting either perpendicular to the fish at the left, or in front of the fish. The dot was moved in discrete jumps at 0.75, 1.5, 3.0, 6.0 or 60.0 Hz at an overall speed of 5 mm s⁻¹ (15.9 degrees (deg) s⁻¹). Each frequency was presented using a dot diameter of 4 mm (12.7 deg). Moreover, 1.5 Hz and 60.0 Hz

stimuli were also presented with dot diameters of 2 mm (6.4 deg) and 8 mm (25.1 deg). Both clockwise and counter-clockwise presentations were shown. The frequency, direction and, if applicable, size were randomly drawn at each stimulus instance. Each stimulus had a duration of 22.6 s and was followed by a 20 s break. A total of 13 stimuli were shown per 10 min recording. Five to nine of these recordings were performed in each fish, leading to an average of four to six presentations of each stimulus. For Fig. 2b, only responses to dots (4 mm diameter) with 1.5 Hz and 60 Hz bout frequency were analysed. For Fig. 2g, again only responses to 4 mm dots were analysed.

**Specificity.** Naturalistic stimulus trajectories consisted of a 4 mm diameter dot (12.7 deg) moving along real trajectories from one of two interacting juvenile zebrafish that were previously recorded[2]. The trajectory was computed as a fish-centric view of the conspecific with respect to a focal fish. To avoid noise in the heading calculation due to tracking jitter, the trajectory was convolved with a normalized hamming kernel (mode: valid, window length: 20). The naturalistic motion sequences were shown for 1 min each (Extended Data Fig. 5b). For the whole-field motion stimulus an image was created by combining random intensities and restricted spatial distributions in Fourier space, matching the size of the moving dot (Extended Data Fig. 5c). The computed image either rotated in discrete jumps of 1.5 Hz or continuously at 60 Hz. In both cases the stimulus took 22.6 s to finish a complete round. All stimuli, 1.5 Hz dot, 60 Hz dot, 1.5 Hz whole-field, 60.0 Hz whole-field and naturalistic dot motion were shown in a pseudo-random order during 6x10 min recordings.

**Kinetic parameters.** Presented 4 mm diameter (12.7 deg) dots moved clockwise on a circular trajectory (18 mm radius). Five speeds were tested using a continuously moving dot: 2.5, 5, 15, 50 and 150 mm s$^{-1}$ (8, 15.9, 47.7, 159.2 and 477.5 deg s$^{-1}$). Five speeds at a bout frequency of 1.5 Hz were tested by increasing the distance the dot moved during each bout. This increased both the average speed and the acceleration during bouts. The following parameters were tested: 1.25 mm s$^{-1}$; 3 m s$^{-2}$, 2.5 mm s$^{-1}$; 6 m s$^{-2}$, 5 mm s$^{-1}$; 12 m s$^{-2}$, 10 mm s$^{-1}$; 24 m s$^{-2}$, and 20 mm s$^{-1}$; 48 m s$^{-2}$ (4 deg s$^{-1}$; 9.5 × 10$^3$ deg s$^{-2}$, 8 deg s$^{-1}$; 19.1 × 10$^3$ deg s$^{-2}$, 15.9 deg s$^{-1}$; 38.2 × 10$^3$ deg s$^{-2}$, 31.8 deg s$^{-1}$; 76.4 × 10$^3$ deg s$^{-2}$ and 63.7 deg s$^{-1}$; 152.8 × 10$^3$ deg s$^{-2}$). Finally, for changing acceleration during each bout, we modelled each bout as a gaussian speed profile and changed the width of the curve. Each stimulus still had an average speed of 5 mm s$^{-1}$ (15.9 deg s$^{-1}$) through a normalization factor. The following peak accelerations were tested: 0.0, 0.02, 0.5, 2.0 and 12.0 m s$^{-2}$ (0, 0.06, 1.6, 6.3 and 38.2 × 10$^3$ deg s$^{-2}$).

**Control stimuli after tectal ablation.** Control stimuli consisted of translational gratings moving caudorostrally with respect to the fish (width, 20 mm; frequency, 0.12 Hz; duration, 20 s) and a looming stimulus (expansion from 0.6 deg to 110 deg in 83 ms, delay 10 s with disk and 20 s without stimulus) centred below the fish. One grating was shown at the beginning, followed by the dot stimuli, another grating and finally the looming stimulus. These recording sessions took 10 min each and were separated by a 1 min break to avoid potential habituation or response suppression due to the looming stimulus.

### Data analysis for two-photon imaging
Suite2P[55] was used for motion correction, ROI detection, cell classification and signal extraction. For the entire analysis, a GCaMP6s time-constant of 7 s was used to accommodate the slow kinetics partially due to the nuclear localization of this sensor. On the basis of a visual inspection of the raw data, a cell diameter of 4–6 px was used. In detail, raw recording files were deinterleaved into separate time series for each plane. An extra motion-correction step was required owing to ripple noise stemming from the resonant mirror: to avoid spurious alignment to the noise pattern, rigid and non-rigid motion correction

was performed on a spatially low-pass filtered time series (Gaussian, sigma = 4). The resulting motion-correction parameters were applied to the raw data. Next, the time series were downsampled fivefold to one volume per second. On the downsampled data, ROIs were detected and fluorescent traces were extracted.

**Thresholding.** Neuron ROIs were thresholded in a two-step process. First, the built-in Suite2p classification algorithm iscell was applied using the default parameters. Second, *iscell*⁺ ROIs that showed a mean Δ$F$/$F$ response to any stimulus above the 95th percentile (Fig. 2)/90th percentile (Fig. 4).

**Mean Δ$F$/$F$ responses.** For each functional ROI, the fluorescent trace was normalized and split into stimulus episodes. Δ$F$/$F$ was computed by using the 5 s before stimulus onset as the baseline. Δ$F$/$F$ temporal responses were averaged across stimulus presentations per stimulus and then averaged over time to receive one value per stimulus.

**BPI.** On the basis of the behavioural tuning curves to bout frequency[2], stimuli were split into bout-like (0.75–3 Hz) and continuous (6–60 Hz) categories, regardless of stimulus size or directionality. BPI was defined as the difference in mean over mean Δ$F$/$F$ to bout-like stimuli and mean over mean Δ$F$/$F$ to continuous stimuli divided by their sum (equation (1)). BPNs were considered all ROIs that scored BPI >0.5, which equates to a threefold higher bout response.

$$\frac{\text{mean } \Delta F/F \text{ (bout)} - \text{mean } \Delta F/F \text{ (continuous)}}{\text{mean } \Delta F/F \text{ (bout)} + \text{mean } \Delta F/F \text{ (continuous)}} \quad (1)$$

**Tuning peaks.** For computing peaks in the tuning of neurons to a variable, mean Δ$F$/$F$ responses were interpolated with a one-dimensional spline (scipy.interpolate.InterpolatedUnivariateSpline, $k$ = 2, second degree) and the location of the maximum was computed.

**Gaussian KDE.** To generate a kernel density estimate of BPNs in anatomical space, BPN coordinates were used to fit a Gaussian Kernel (sklearn.neighbors.KernelDensity(*, bandwidth = 10 (14 for 7 d.p.f.), algorithm=‘auto’, kernel=‘gaussian’, metric=‘euclidean’). In detail, the brain was divided along the rostrocaudal axis and, for each hemisphere, a separate kernel was fitted with the contained BPNs. The resulting two kernels were used to generate probability density fields of each hemisphere, which were then merged again. The resulting density was thresholded so that only voxels within the brain itself had values > 0 and all voxels in the volume surrounding the brain equalled 0. Probability values were then normalized so that the sum would result in the total number of BPNs. To draw contours of areas with certain threshold BPN density, the KDE volume was binarized so that all voxels above threshold equalled 1. Of the resulting binarized volume a two-dimensional maximum intensity projection was computed for each orthogonal anatomical axis and a contour-finding algorithm (skimage.measure.find_contours) was applied to the two-dimensional projection.

### Definition of the larval DT
The outline of the larval thalamus proper was refined with expert help of M. Wullimann (LMU). The refinement was based on extensive analysis of gene expression[52]. The *elavl3* reference stain was used to identify the diencephalic regions. Proliferative cells, however, which are abundant in the anterior DT at the larval stage, are not labelled by *elavl3*. The neurogenin line was used to indicate the early glutamatergic cells belonging to DT. Neurogenin is absent in the prethalamus (VT). The VT/DT boundary was further defined using *gad1b* and *dlx4*, which label late and early GABAergic cells, respectively. GABAergic cells are mainly found in VT, although the intercalated nucleus and the anterior nucleus of DT may contain some *gad1b* positive cells. The pretectum/DT

boundary was defined using *gad1b* and *th*. The latter marks dopamine cells present in the pretectum.

### EM and segmentation of mapzebrain regions

A detailed description of the EM dataset and region mapping is published elsewhere[35]. In brief, the Serial Block Face Scanning EM dataset was of a 5 d.p.f. larval zebrafish imaged at a resolution of $14 \times 14 \times 25$ nm. A diffeomorphic mapping between the mapzebrain light-microscopy brain reference coordinate system and the EM coordinate system generated by the dipy (https://dipy.org/) Python library was used to overlay mapzebrain (https://mapzebrain.org) region annotations over the EM data. Registration accuracy was reviewed for different brain regions with an alignment error of maximal ~5 µm (midbrain) to ~20 µm (hindbrain). We applied flood-filling networks for an automated reconstruction of all neurons[56] within the whole-brain EM dataset[35]. To correct for split and merge errors of the segmentation, we used the Knossos application (www.knossos.app). Proof-reading of single pBPN-DT cells started at the cell body location and ended when all branches were completely traced. Growth cones defined premature neurons. Proof-reading of partner cells started at the synapse and was again performed until the whole cell was completed. Synapses have been automatically segmented using the SyConn v2 pipeline[35]. Input to pBPN-DT cells and tectal PVPNs was quantified by randomly selecting ten incoming synapses per cell and tracing input partner cells, until their cell bodies were identified.

### Nitroreductase ablations

To chemogenetically ablate neurons, fish expressing nitroreductase (NTR) were treated with 7.5 mM metronidazole (MTZ) in Danieau's solution with 1 ml l$^{-1}$ dimethylsulfoxid (DMSO) for larvae, or 7.5 mM MTZ in fish facility water with 2 ml l$^{-1}$ DMSO for juvenile fish, respectively. Control fish were only treated with the respective DMSO concentration in the absence of MTZ. For *SAGFF(lf)81c* ablations, the canonical *UAS-E1B:NTR-mCherry(c264)* nitroreductase was used[47].

Transgenic animals (RFP$^+$) and control sibling fish (RFP$^-$) were incubated in MTZ + DMSO solution for 16–24 h overnight. For the experiment shown in Extended Data Fig. 7b,e, we additionally incubated RFP$^+$ and RFP$^-$ animals in DMSO only. Animals recovered for 16–24 h in system water before starting imaging or behaviour experiments.

In larvae, *s1026tEt* drives strong expression in the DT, dorsal VT, ventral pretectum and ventral telencephalon, and additional background expression in the heart and trunk musculature[31,48]. We found that MTZ mediated ablation of *s1026tEt*, *UAS-E1B:NTR-mCherry(c264)* double transgenic fish was lethal in 100% of animals. We therefore used a nitroreductase transgene of which the background muscle expression was suppressed by a 3'UTR: *UAS:BGi-epNTR-TagRFPT-utr.zb3*[57]. To overcome strong variegation of the existing allele (*y362*) in *s1026tEt* cells, we created new alleles via Tol2 mediated transgenesis. We injected *UAS:BGi-epNTR-TagRFPT-utr.zb3* together with *Tol2* mRNA into the TLN *s1026tEt* background. We outcrossed individual founders to TLN and raised transgenic RFP$^+$ offspring and control RFP$^-$ siblings. At 19 d.p.f., we selected transgenic offspring of one founder for homogeneous RFP signal in the DT. This founder established the allele *mpn420* of which the expression is largely confined to the DT (Extended Data Fig. 8). Half of transgenic and control animals were randomly assigned to MTZ + DMSO versus DMSO-only treatment. We observed no lethality after ablation in this line. Analogous, non-transgenic siblings were also split into two groups for MTZ + DMSO versus DMSO-only treatment. MTZ had no detectable effect on non-transgenic animals and we subsequently pooled all three control conditions.

### Laser ablations

Juvenile (20–23 d.p.f.) *elavl3:H2B-GCaMP6s* transgenic fish were embedded as described above, anesthetized with Tricaine and placed under a two-photon laser scanning microscope (Femtonics). The DT-BPN region within each brain was visually identified by the experimenter. A 20 µm ROI was specified on the DT-BPN region of each brain hemisphere and scanned with a 800 nm/400 mW laser beam for 30 ms. After each scan, one image was captured to observe the resulting damage and potential off-target effects. This procedure was repeated $7 \pm 3$ times until no more nuclei could be observed in the target region that could be targeted without blood vessel damage. Fish were removed from the embedding and anaesthesia immediately after ablation. One fish that did not start swimming within a few minutes was excluded from subsequent behaviour testing and analysis. Fish were allowed to recover for at least 16 h before they were tested in a shoaling assay as described above.

### Social isolation

Individual eggs from *elav:H2B-GCaMP6s* incrosses were placed into small petri dishes at 0 d.p.f. The side walls of the dishes were taped to prevent visual contact between dishes. For controls, 15 eggs were placed in one small petri dish. Larvae were imaged at 7 d.p.f. As no brain stacks were acquired for these animals, anatomical DT masks were drawn for each fish manually. The response threshold for data analysis was adjusted to 50% due to a lower number of recorded neurons in single-plane imaging.

### Statistical analysis

All analyses were performed in Python, using NumPy, Scipy, MatplotLib, Suite2p, Pandas and Scikit-learn. All statistical details are described in the figure legends and the Methods. All tests were two-tailed, unless noted otherwise. Error bars represent 1 s.d., unless noted otherwise. *N* denotes the number of animals, unless noted otherwise.

### Data collection software

The following data collection software were used: Bonsai (v.2.4.1); Leica LAS X (v.3.5.7); and ScanImage (v.5.6).

### Data analysis software

The following data analysis software was used: Python (v.3.9) with NumPy (v.1.21.0), Scipy (v.1.7.0), MatplotLib (v.3.4.2), Pandas (v.1.3.0) and additional packages (full python environments are available with our code on bitbucket); Ants (v.1.9); Suite2p (v.0.9.3); and ImageJ (v.1.53c).

### Reporting summary

Further information on research design is available in the Nature Research Reporting Summary linked to this paper.

## Data availability

All data to evaluate the conclusions in the paper and to reproduce the analysis are provided in the Article or made publicly available. Raw HCR data, two-photon time series for individual neurons and behaviour tracking data are available at Edmond (https://doi.org/10.17617/3.2QCFQP)[58]. The EM stack will be publicly available in a companion paper[35]. Source data are provided with this paper.

## Code availability

Python scripts to recapitulate our data analysis are available at bitbucket (https://bitbucket.org/mpinbaierlab/kappeletal2022).

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

**Acknowledgements** We thank E. Laurell and E. Kuehn for generating HCR stainings and for imaging marker lines for the mapzebrain atlas. Funding was provided by the Max Planck Society. J.K. was supported by a Boehringer Ingelheim Fonds graduate fellowship. J.L. was supported by a NARSAD Young Investigator Award. I.S. was supported by an Alexander von Humboldt foundation research fellowship.

**Author contributions** J.M.K. and K.S. performed two-photon imaging experiments. J.M.K. performed laser ablations, registration of two-photon and HCR data. M.J. segmented the EM volume with FFNs. D.F. proofread and traced presegmented EM data. I.S. performed HCR staining and imaging. F.S. generated the EM data and performed EM brain area registration. S.S. performed pilot HCR in situ experiments in juvenile fish. J.C.D. helped with the two-photon hardware and remote focusing, and advised on microscopy and analysis. J.L. performed behaviour experiments, transgenesis and brain area segmentation. J.M.K., K.S., D.F. and J.L. analysed the data. J.M.K., K.S., D.F., H.B. and J.L. interpreted the data. J.M.K., D.F., H.B. and J.L. wrote the paper with input from all of the authors. All of the authors reviewed and edited the manuscript. H.B. and J.L. supervised the project.

**Funding** Open access funding provided by Max Planck Society.

**Competing interests** The authors declare no competing interests.

**Additional information**
**Correspondence and requests for materials** should be addressed to Herwig Baier or Johannes Larsch.

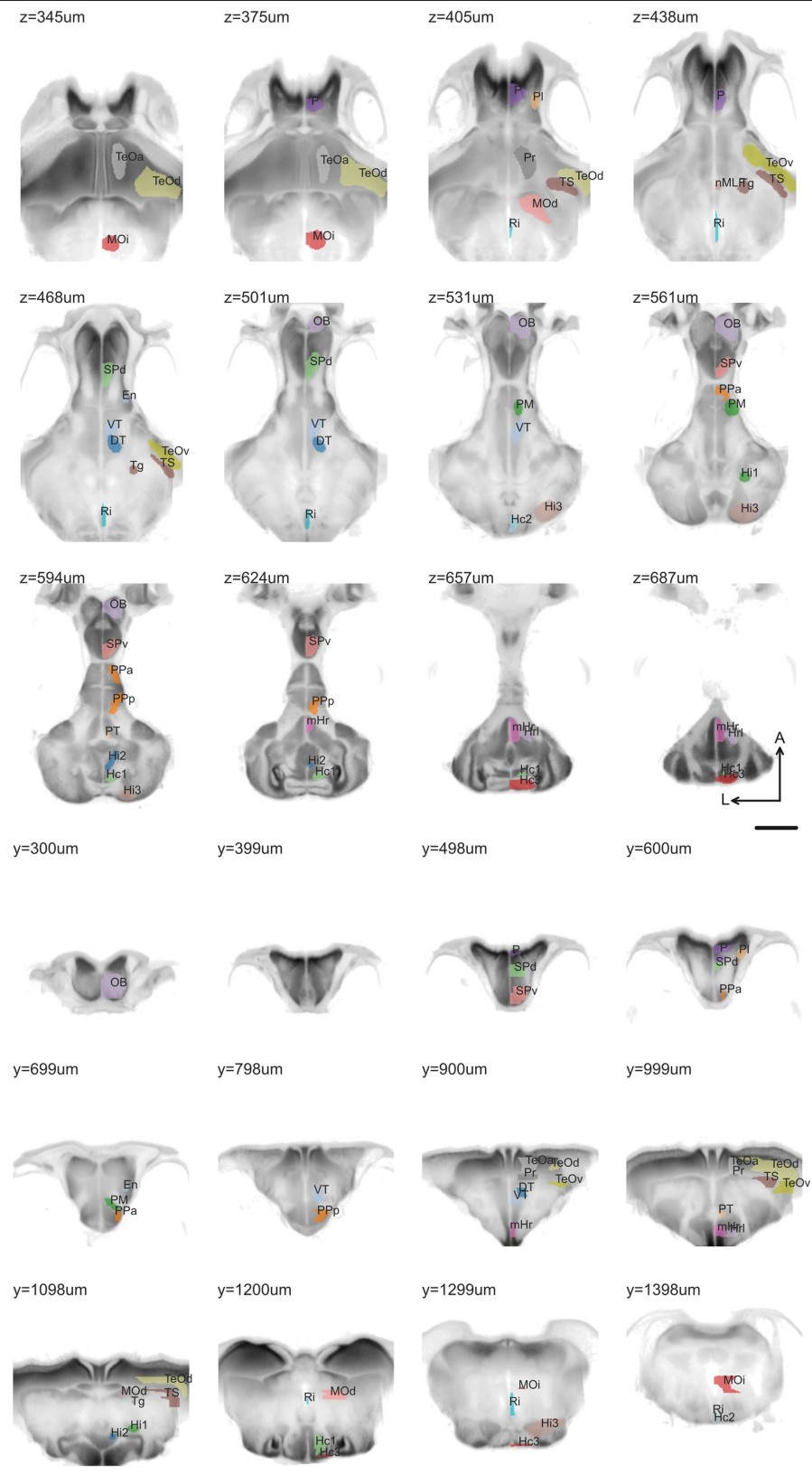

**Extended Data Fig. 1 | Overview of all manually segmented *c-fos* activity clusters.** Horizontal and coronal slices showing manually segmented *c-fos* activity clusters overlaid on the mean registered *elavl3* signal across all 28 animals. For visualization purposes, the *elavl3* signal was non-linearly transformed using a gamma adjustment of 0.5. Top three rows are horizontal sections, bottom three rows are coronal sections. Scale bar: 200 μm.

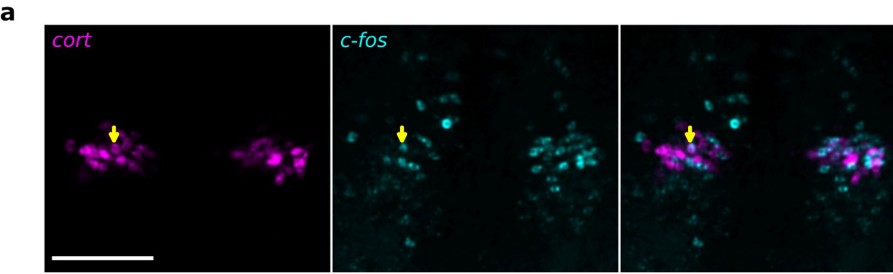

**a**

**b**

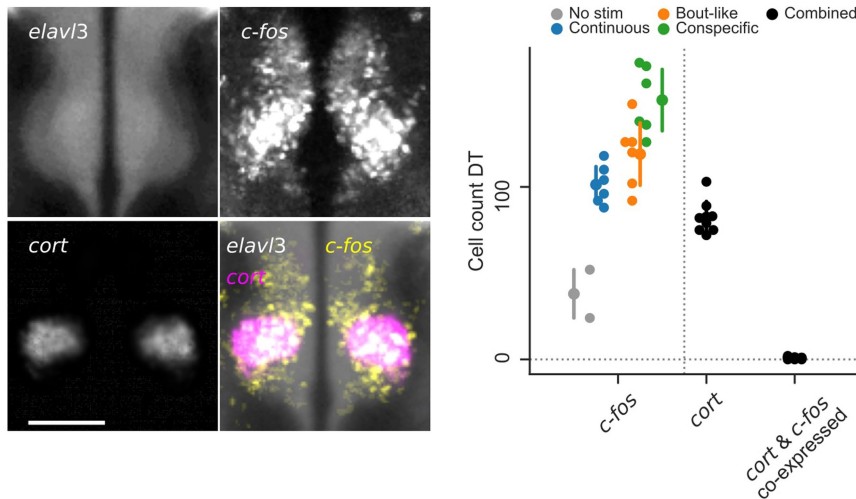

**c**

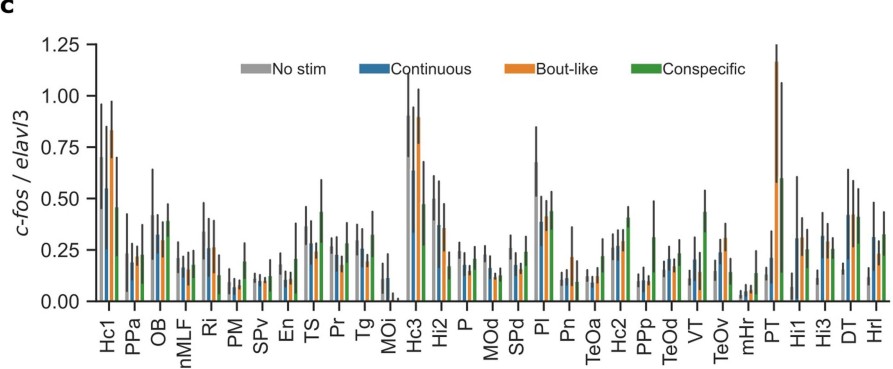

**Extended Data Fig. 2 | Quantification of *c-fos* and *cort* HCR labelling.**
**a**, Dorsal thalamic plane showing HCR labelling of *c-fos* and *cort* in one animal after interaction with a bout-like motion dot stimulus. Expression occurs in the same regional cluster (DT) but only one cell expresses both markers (arrowhead). Representative for 6 examined animals. **b**, Left: Expression of *cort* and *c-fos* induction by shoaling stimuli localize to the same area. *elavl3*, *cort* and *c-fos* were co-labelled in the same animals. A single horizontal imaging plane at the centre of the DT cluster is shown. *elavl3* and *cort* channels are mean intensity, *c-fos* channel is maximum intensity over all 28 animals. Right: *c-fos*⁺ and *cort*⁺ cells were counted in the dorsal thalamus of N = 2,6,6,6,8,8 animals for no stim, continuous, bout like, conspecific, *cort*, and *cort* & *c-fos* groups, respectively. Data are mean ± 1SD. **c**, Normalized bulk *c-fos* signal for each activity cluster. Bars represent median±1SD, N = 6,8,6,8 animals for 'No stim', 'Continuous', 'Bout-like', 'Conspecific', respectively. Individual data points are available in the figure source data. Scale bars: 100 µm.

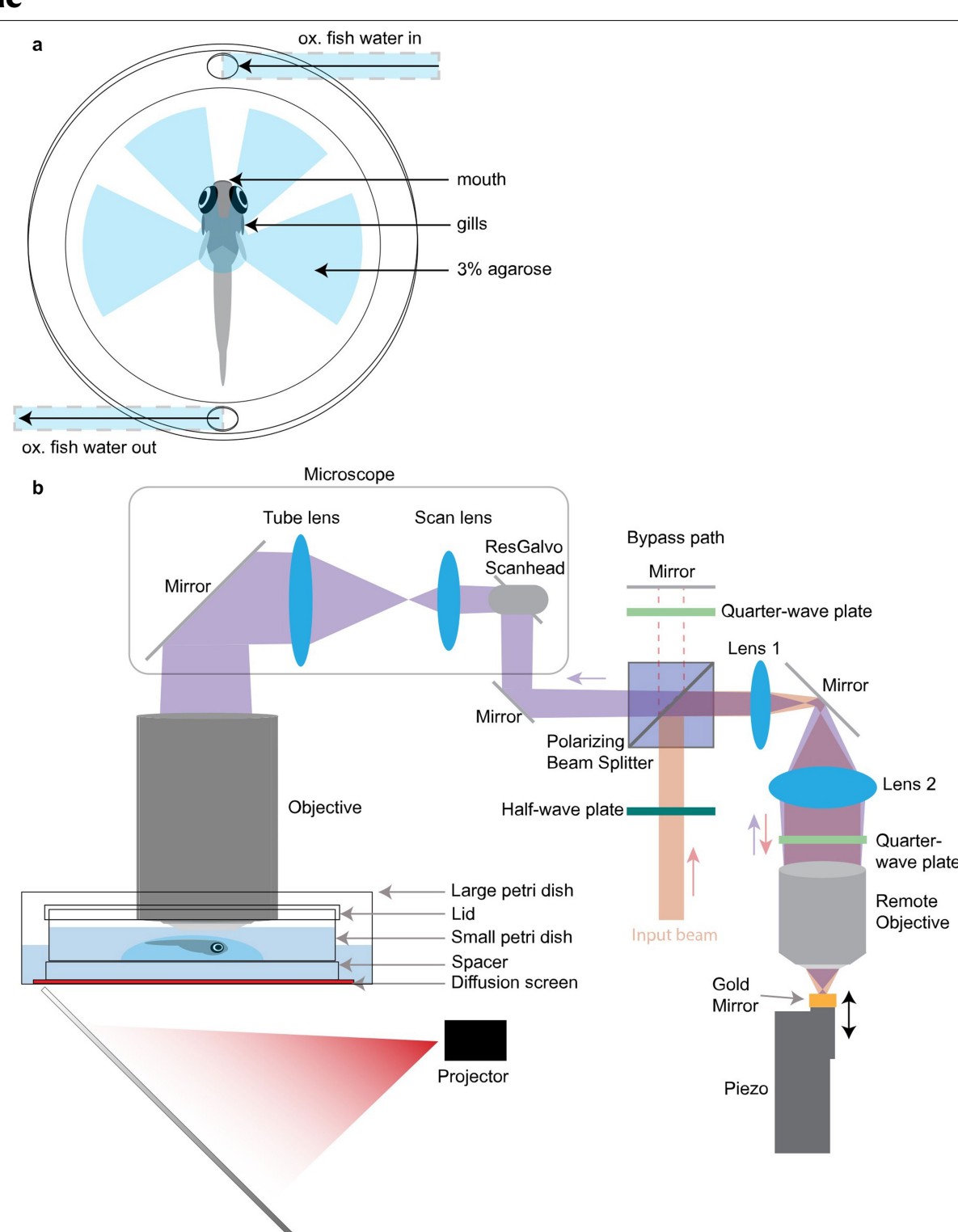

**Extended Data Fig. 3** | See next page for caption.

**Extended Data Fig. 3 | Schematic of the setup for calcium imaging.**
**a**, Top view of the embedding preparation for 2-photon imaging of juvenile zebrafish. To enable active respiration, agarose columns are cut out in front of the mouth and gills. The tail is also freed to improve oxygen uptake through the skin. Oxygenated water in the imaging chamber is constantly renewed with a peristaltic pump. **b**, Side view of the preparation and remote focusing system. The imaging chamber, consisting of a small petri dish, is placed in a large petri dish filled with water. Diffusive paper serving as a screen and a small spacer are placed between the large and small petri dish. The large petri dish is placed on a custom-made sample holder. A cold mirror is placed under the preparation to reflect projector images onto the screen. The input beam to the remote focusing system (red), passes through a half-wave plate and is reflected by a polarizing beam splitter. The beam is enlarged by two lenses, passes through a quarter-wave plate, and is focused by an objective onto a mirror mounted to a custom piezo stage. The piezo moves the mirror and thus adjusts the effective focal distance of the reflected beam, which ultimately changes the collimation of the beam at the main objective, changing the focus. The second pass through the quarter-wave plate on the return trip results in a change of polarization compared with the input beam, so the reflected beam now continues straight through the polarizing beam splitter, reaching the microscope. To bypass the remote focusing path, the input half-wave plate can be rotated so the input beam instead passes through the polarizing beam splitter, hits a mirror and passes through a quarter-wave plate twice, and then is reflected into the microscope. The detection path is standard and is not depicted.

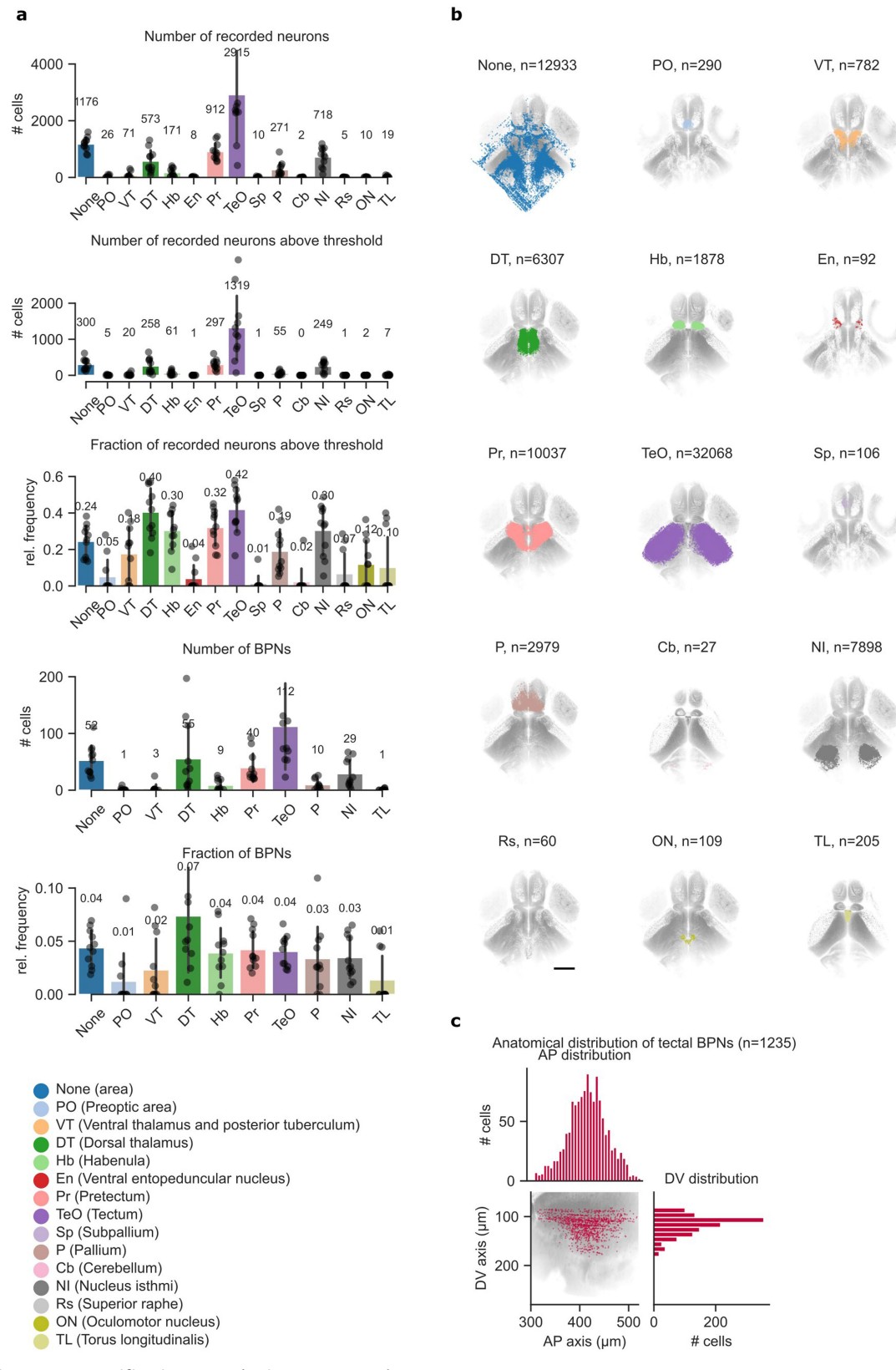

**Extended Data Fig. 4 | BPN quantification across brain areas. a**, Bar plots showing average numbers and fractions of neurons per fish per brain area. Bars indicate mean across animals, error bars represent 1SD. Dots show mean of individual fish. Top: Number of recorded neurons per brain area. Second: Number of recorded neurons above response threshold, see methods. Middle: Fraction of recorded neurons that surpassed response threshold per brain area. Fourth: Average number of BPNs per brain area. Bottom: Fraction of recorded neurons that were classified as BPNs per brain area. N = 11 animals. **b**, Anatomical location of recorded neurons in each brain area. Dots represent individual neurons. **c**, Distribution of tectal BPNs along anterior-posterior (AP) and dorso-ventral (DV) axis. Scale bar: 200 μm.

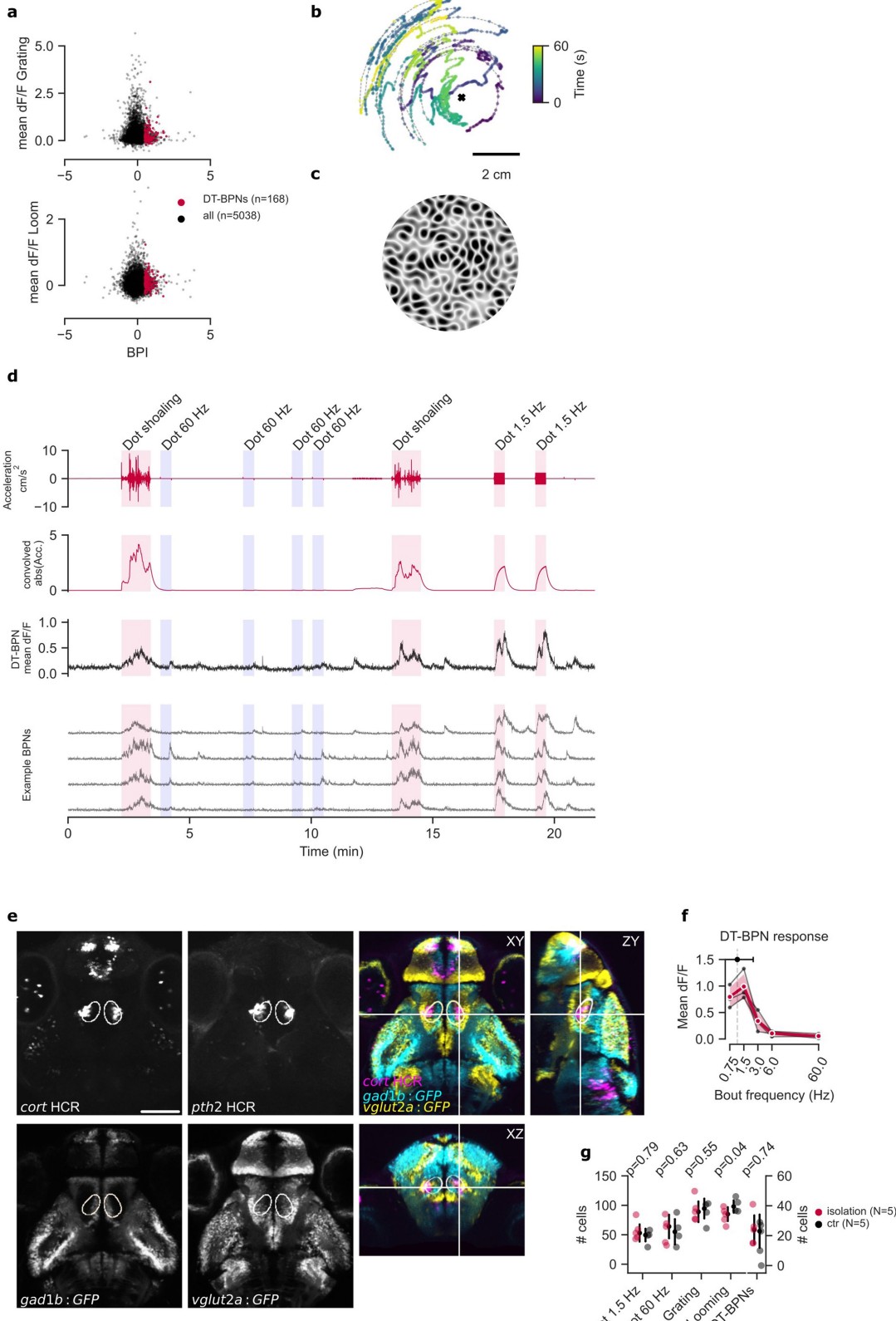

**Extended Data Fig. 5** | See next page for caption.

**Extended Data Fig. 5 | BPN response specificity in juvenile and larval animals. a**, Responses of DT-BPNs (n = 56 ± 51) and all other recorded neurons above threshold (n = 1679 ± 1065) to looming and translational grating stimuli from 3 animals where these stimuli were included in the protocol. DT-BPNs are not strongly activated by either of the two control stimuli. **b**, Dot shoaling stimulus. Dot position recapitulates the location of a conspecific relative to a focal fish facing up. Cross marks location of the focal fish. **c**, Whole-field motion stimulus used in Fig. 2i. **d**, DT-BPNs respond to dot shoaling stimulus. Top red trace shows instantaneous dot stimulus acceleration, bottom red trace the same time series convolved with GCaMP6s kernel (see methods). Dot shoaling, 1.5 Hz and 60 Hz stimuli were shown in pseudo-random order as indicated. Median trace represents n = 84 DT-BPNs from one fish, including four representative neurons shown below. **e**, Expression of *cort*, *pth2*, *gad1b* and *vglut2a* defines the location of the larval BPN KDE as dorsal thalamus. *gad1b* positive 'stripe' of cells near the midline marks the dorsal edge of VT.

Left four panels show single planes. Right shows merged orthogonal views. Each channel shows mean expression over multiple individual fish registered to the mapzebrain atlas. *cort* and *pth2* are HCR labels, mean of N = 3 animals each. *gad1b* and *vglut2a* are Gal4 enhancer trap lines driving expression of GFP, N = 5 animals each. Also see Video S1. **f**, Larval DT-BPN (n = 51 ± 14 neurons per fish, N = 4 animals) tuning curve to stimulus frequencies from 0.75 to 60 Hz shown in red. Mean tuning peak of individual neurons (shown above) was 1.1 Hz ± 1.3 Hz. Black lines represent means of individual animals. Data from a subset of all animals in Fig. 2j with number of recorded DT-BPNs > 30. **g**, Number of neurons in DT responding to different visual stimuli of larvae raised in social isolation compared to group-raised larvae (N = 5 animals per group). Number of responsive neurons to bout-like motion as well as the number of DT-BPNs are not significantly different in socially isolated animals. Error bars: 1SD. P-value: two-tailed student's t-test, no corrections. Scale bars: b, 2 cm; e, 100 μm.

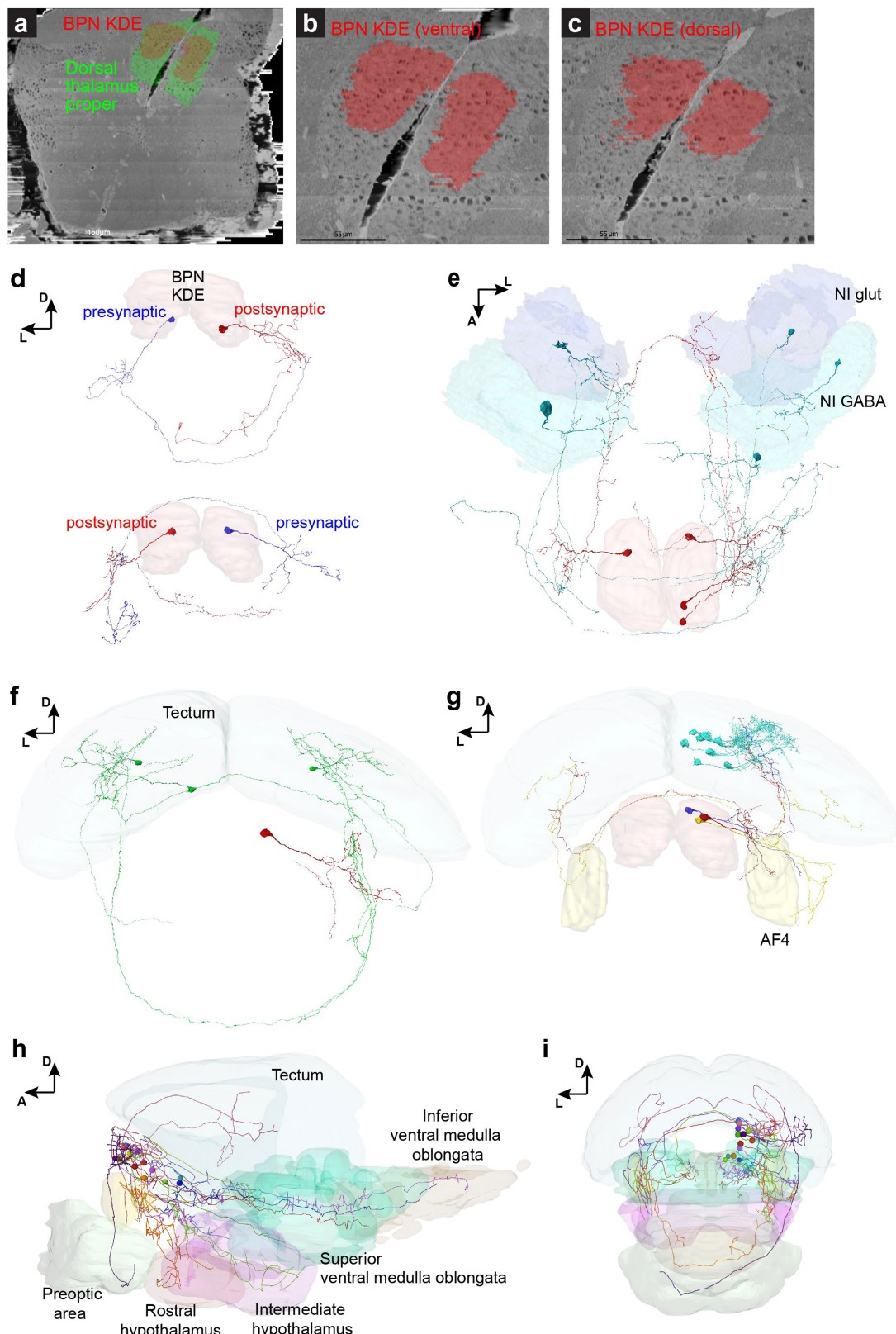

**Extended Data Fig. 6 | Representative example neurons and brain areas in the EM dataset. a-c**, Registration of mapzebrain regions to the EM dataset. Shown are top-views of the kernel density estimation for bout-preference neurons (red) inside the dorsal thalamus proper (green, a) and at different coronal planes (b, c). N = 1 EM stack. **d**, Frontal views of the BPN KDE in EM reconstructions showing two examples (top and bottom) for synaptically connected putative BPN partners across brain hemispheres. **e**, Dorsal view of three putative BPNs (red) and their presynaptic partners in the nucleus isthmi

(glutamatergic and GABAergic domains are annotated). **f**, A single putative BPN (red) receives ipsi- and contralateral synaptic input from at least three identified tectal PVPNs (green). **g**, Three examples for putative BPNs (red, orange, purple) with axonal projections to the ipsi- and contralateral tectum. Eight tectal PVINs (cyan), which are postsynaptic to the red putative BPN are shown. **h-i**, Mapzebrain atlas showing single cells in the BPN KDE and their targeted brain regions in lateral (h) and frontal (i) views. AF4 (yellow) is shown as a reference. Scale bars: a, 150 μm; b, c, 55 μm.

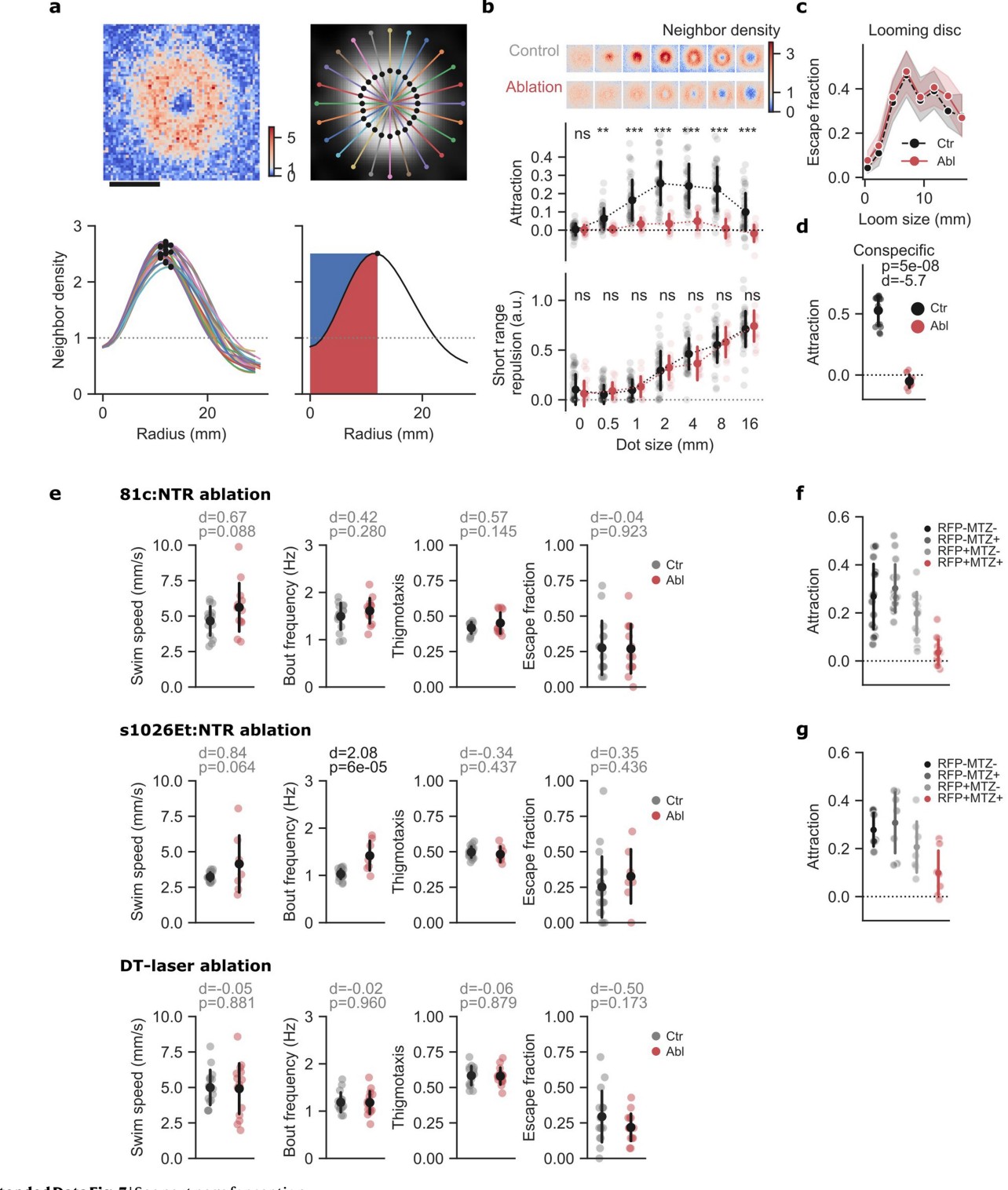

**Extended Data Fig. 7 | Additional behavioural analysis of ablated animals. a**, Definition of short-range repulsion. Top left shows a representative neighbour map for one of 45 similar control animals interacting with a dot of 4 mm diameter. Colour map as in Fig. 4e. Top right and bottom left show 24 radial line scans of smoothed neighbour density. Each scan begins at the centre of the map. Black dots label the maximum of each scan. Bottom right shows the mean of all line scans in black. Repulsion is quantified as the area above the mean line, left of the peak (blue shading). This defines the reduction of attraction at the centre of the map. Dotted line in bottom panels indicates baseline neighbour density (random distribution). **b**, Attraction and neighbour density are strongly reduced in *81c:NTR* ablated animals. Short-range repulsion is intact. N = 15 ablated, 45 control animals. Data represent individual animals and mean (attraction) or median (repulsion). Neighbour maps show mean probability of finding the stimulus in space with the focal animal at the centre of the map, heading up. Values represent ratios, relative to time-shuffled data. Each map is 60 x 60 mm. ***: p < 0.001/7, **: p < 0.01/7, ns: p > 0.05/7. Two-sided student's t-test for attraction, Mann-Whitney U test for repulsion, alpha values Bonferroni corrected for 7 comparisons. See figure source data for individual p-values. **c**, Loom induced startle responses are intact in *81c:NTR* ablated animals. Same animals as in Fig. 4c, d. Data represents mean, shading represents 1SD. N = 13 ablated, N = 15 control animals. **d**, Shoaling with real conspecifics is disrupted in *81c:NTR* ablated animals. Fish as in Fig. 4c, d +1 additional animal per group tested in pairs. N = 14 ablated animals, tested as 7 pairs, N = 16 control animals, tested as 8 pairs. **e**, Quantification of control behaviours in animals shown in Fig. 4c, d, i, and k. Refer to legend of Fig. 4 and figure source data for additional statistical information. **f**, Attraction shown separately for 81c:NTR-RFP ablation +/− RFP and MTZ +/− control groups, same data as 2 mm dots in b. RFP: red fluorescent protein indicates expression/ no expression of nitroreductase, respectively. MTZ: Metronidazole incubation/ DMSO control, respectively. N = 15 animals in each group. **g**, Attraction shown separately for s1026tEt:NTR-RFP ablation +/− RFP and MTZ +/− control groups, same data as in e. N = 7 animals in each group. Data in d-g show individual animals or pairs and mean. Error bars are 1SD. p: Two-sided student's t-tests, uncorrected. d: Cohen's *d*. Scale bar: 20 mm.

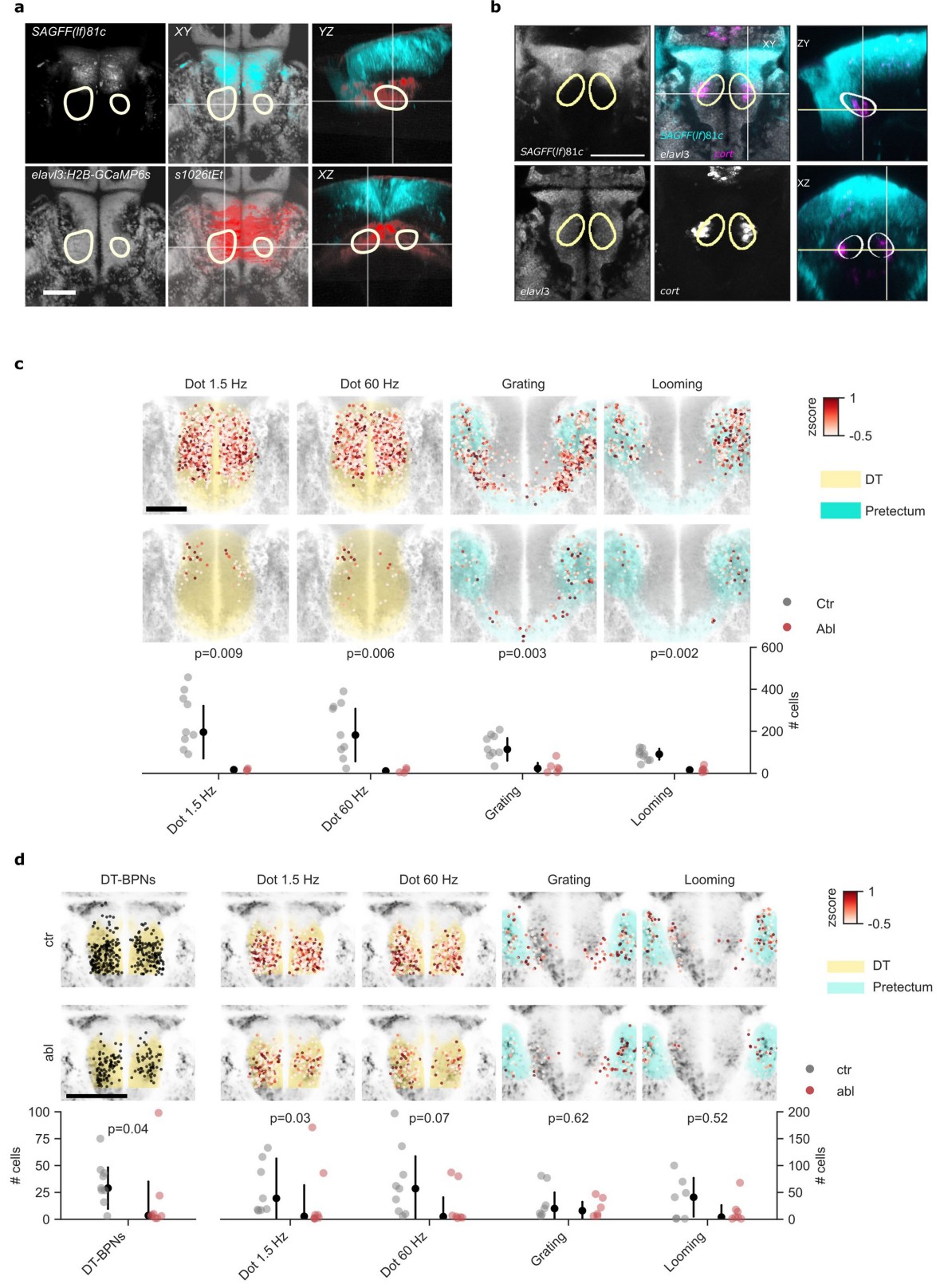

**Extended Data Fig. 8 |** See next page for caption.

**Extended Data Fig. 8 | Evoked neuronal activity in *SAGFF(lf)81c* ablated animals. a**, Juvenile expression of *SAGFF(lf)81c:Gal4*, *UAS:NTR-mCherry* and *s1026tEt, UAS:NTR-mCherry* relative to BPN KDE (yellow outline). Markers were imaged in separate 21 dpf fish and registered to the juvenile standard brain. *SAGFF(lf)81c*: average of 3 animals. *s1026tEt:* average of 2 animals. *elavl3:H2B-GCaMP6s:* mapzebrain reference channel. Merged view is shown in 3 orthogonal axes. White crosshairs indicate the orthogonal planes. **b**, Larval expression of *SAGFF(lf)81c:Gal4*, *UAS:NTR-mCherry* relative to BPN KDE (yellow outline) and *cort* HCR. Markers were imaged in separate 5–7 dpf fish and registered to the mapzebrain standard brain. *81c*: average of 4 animals *SAGFF(lf)81c:Gal4, UAS:NTR-mCherry. elavl3*: mapzebrain reference channel. *cort*: average of 3 animals, HCR label. Merged view is shown in 3 orthogonal planes. White crosshairs indicate the orthogonal planes. Also see Video S3. **c**, Additional analysis of Fig. 4e–h. Number of neurons with mean dF/F above the 90th percentile in DT for bout-like dot motion, continuous dot motion, as well as in pretectum for translational gratings and looming stimuli of *81c:NTR* ablated and control animals, respectively. Ablated animals show significantly less responding neurons for all stimuli compared to controls. Individual data points denote animals. Color code of individual neurons is z-scored mean dF/F per stimulus. N = 7 ablated, 9 control animals. **d**, Same experiment as in Fig. 4e–h, and Extended Data Fig. 8c, but in 7dpf larvae. Here, significant reduction of responding neurons in *81c:NTR* ablated animals is only observed for DT-BPNs and bout-like dot motion in DT, whereas continuous dot motion, translational gratings and looming do not evoke significantly less responses in the pretectum of ablated animals. N = 8 ablated, 9 control animals. Error bars: 1SD. Scale bars: a, b, 100 μm; d. p-values in c,d result from two-sided Mann-Whitney U test (uncorrected).

# Reporting Summary

## Statistics

For all statistical analyses, confirm that the following items are present in the figure legend, table legend, main text, or Methods section.

| n/a | Confirmed | |
|---|---|---|
| ☐ | ☒ | The exact sample size (*n*) for each experimental group/condition, given as a discrete number and unit of measurement |
| ☐ | ☒ | A statement on whether measurements were taken from distinct samples or whether the same sample was measured repeatedly |
| ☐ | ☒ | The statistical test(s) used AND whether they are one- or two-sided *Only common tests should be described solely by name; describe more complex techniques in the Methods section.* |
| ☒ | ☐ | A description of all covariates tested |
| ☐ | ☒ | A description of any assumptions or corrections, such as tests of normality and adjustment for multiple comparisons |
| ☐ | ☒ | A full description of the statistical parameters including central tendency (e.g. means) or other basic estimates (e.g. regression coefficient) AND variation (e.g. standard deviation) or associated estimates of uncertainty (e.g. confidence intervals) |
| ☐ | ☒ | For null hypothesis testing, the test statistic (e.g. *F*, *t*, *r*) with confidence intervals, effect sizes, degrees of freedom and *P* value noted *Give P values as exact values whenever suitable.* |
| ☒ | ☐ | For Bayesian analysis, information on the choice of priors and Markov chain Monte Carlo settings |
| ☒ | ☐ | For hierarchical and complex designs, identification of the appropriate level for tests and full reporting of outcomes |
| ☐ | ☒ | Estimates of effect sizes (e.g. Cohen's *d*, Pearson's *r*), indicating how they were calculated |

*Our web collection on statistics for biologists contains articles on many of the points above.*

## Software and code

Policy information about availability of computer code

| Data collection | Bonsai 2.4.1 Leica LAS X 3.5.7 ScanImage 5.6 |
|---|---|
| Data analysis | Python 3.9 with NumPy 1.21.0, Scipy 1.7.0, MatplotLib 3.4.2, Pandas 1.3.0, and additional packages. (Full python environments are available with our custom scripts on bitbucket). Ants 1.9 Suite2p 0.9.3 ImageJ 1.53c |

For manuscripts utilizing custom algorithms or software that are central to the research but not yet described in published literature, software must be made available to editors and reviewers. We strongly encourage code deposition in a community repository (e.g. GitHub). See the Nature Portfolio guidelines for submitting code & software for further information.

## Data

Policy information about availability of data

All manuscripts must include a data availability statement. This statement should provide the following information, where applicable:
- Accession codes, unique identifiers, or web links for publicly available datasets
- A description of any restrictions on data availability
- For clinical datasets or third party data, please ensure that the statement adheres to our policy

All data to evaluate the conclusions in the paper and to reproduce the analysis are present in the paper or made publicly available. Raw HCR data, 2-photon time

March 2021

series for individual neurons and behavior tracking data are available on Edmond: https://doi.org/10.17617/3.2QCFQP. The EM stack will be publicly available with a companion paper (Svara et al.).

# Field-specific reporting

Please select the one below that is the best fit for your research. If you are not sure, read the appropriate sections before making your selection.

☒ Life sciences ☐ Behavioural & social sciences ☐ Ecological, evolutionary & environmental sciences

For a reference copy of the document with all sections, see nature.com/documents/nr-reporting-summary-flat.pdf

# Life sciences study design

All studies must disclose on these points even when the disclosure is negative.

| | |
|---|---|
| Sample size | The sample size was not pre-computed. We chose sample sizes typical in the field, expecting large effect sizes (greater than 1SD) and refrained from strong conclusions based on the absence of significance, which would be limited by modest sample size. |
| Data exclusions | No individual data points were excluded from the analysis. Animals damaged during the embedding, during imaging, or during ablation were not analyzed further as described in the methods section. |
| Replication | All findings described in the manuscript were analyzed and found to be consistent across multiple animals, except the EM stack, which only exists for N=1 animal. Animal numbers are indicated in each figure legend. |
| Randomization | For c-fos mapping, animals were randomly assigned to stimulus groups. Test and control groups for MTZ treatment were assigned randomly. |
| Blinding | The experimenters were not blinded to experimental conditions or genotype of the animals. We applied a standardized, fully automated analysis to data from all animals. |

# Reporting for specific materials, systems and methods

We require information from authors about some types of materials, experimental systems and methods used in many studies. Here, indicate whether each material, system or method listed is relevant to your study. If you are not sure if a list item applies to your research, read the appropriate section before selecting a response.

| Materials & experimental systems | | Methods | |
|---|---|---|---|
| **n/a** | **Involved in the study** | **n/a** | **Involved in the study** |
| ☒ | ☐ Antibodies | ☒ | ☐ ChIP-seq |
| ☒ | ☐ Eukaryotic cell lines | ☒ | ☐ Flow cytometry |
| ☒ | ☐ Palaeontology and archaeology | ☒ | ☐ MRI-based neuroimaging |
| ☐ | ☒ Animals and other organisms | | |
| ☒ | ☐ Human research participants | | |
| ☒ | ☐ Clinical data | | |
| ☒ | ☐ Dual use research of concern | | |

## Animals and other organisms

Policy information about studies involving animals; ARRIVE guidelines recommended for reporting animal research

| | |
|---|---|
| Laboratory animals | Zebrafish (Danio rerio), age: 7-22 days. Transgenic lines are specified in the methods section. The sex of zebrafish is likely determined but not yet differentiated at this stage and was thus not a factor for choosing experimental animals. |
| Wild animals | No wild animals were used in this study. |
| Field-collected samples | No field collected samples were used in this study. |
| Ethics oversight | All animal experiments were performed under the regulations of the Max Planck Society and the regional government of Upper Bavaria (Regierung von Oberbayern), approved protocols: ROB-55.2Vet-2532.Vet 03-15-16, ROB-55.2Vet-2532.Vet 02-16-31, and ROB55.2Vet-2532.Vet 02-16-122. |

Note that full information on the approval of the study protocol must also be provided in the manuscript.

