## [Peer Review File · Nature]

Manuscript Title: Visual recognition of social signals by a tecto thalamic neural circuit

Reviewer Comments & Author Rebuttals

Reviewer Reports on the Initial Version:

Referees' comments:

Referee #1 (Remarks to the Author):

In this manuscript, the authors provide evidence for a tectal-thalamic circuit that is important for shoaling/social affiliation in zebrafish. While many genes/neuromodulators have been associated with shoaling/social affiliation in teleosts, a neural circuit for these behaviors is not yet characterized, making this paper an exciting step forward. The authors use a number of different approaches to tackle this questions, including in vivo imaging, neural activity mapping, chemogenetics, and utilizing an electron microscopy-derived connectome map of the larval zebrafish brain. The data is robust, although in its current form, it is sometime unclear why the authors made certain choices in which brain regions/cells to investigate.

General comments:

1. Social behavior network: I appreciate that the authors are setting their results within the wider literature of brain networks that govern social behavior across vertebrates – the Social Behavior Network originally proposed by Newman and composed of the medial amygdala, lateral septum, preoptic area, anterior hypothalamus, ventromedial hypothalamus, and periaqueductal/central gray. However, I think that the application of this well-established network referencing very specific brain regions in vertebrates to non-analogous (?) brain regions in teleosts that are more active in real/virtual conspecifics is misleading. For example, line 96 states that the authors have identified a social behavior network whose activity is modulated by real and virtual conspecifics. However, based on my interpretation of Figure 1e, this could include PT, mHr, and Hi1, as the other brain regions are also activated by continuous (non-fish-like) motion. The authors should consider stating that brain regions PT, mHr, and Hi1 are analogous to in other brains regions in the social behavior network, considering adopting a different name, or propose expanding the current network to include these regions in the context of shoaling. Similarly, the PVPN -> DT-BPN -> POA pathway seems to be the only connection to the classical “Social Behavior Network”, unless I am misinterpreting fish brain regions. I do not dispute that these DT neurons project to places important for social behavior but caution the use of this phrase as something that may be confusing to readers familiar with established literature on the topic.
2. Looking at the data in 1e, group 5, it is unclear to me why the authors decided to focus on the dorsal thalamus and not the other brain regions that were activated in bout-like and conspecific situations, but not with the continuous stimuli.
3. It is interesting that there is activation of tectal neurons only with real conspecifics (Fig 1e) and not with the other visual stimuli. Do the authors have ideas for what this means for visual processing of real versus virtual stimuli? The chemoablations of these neurons do influence the affiliative behavior, so perhaps there is only a tiny subset that encodes conspecific acceleration/speed?

4. The authors propose a PVPN -> DT-BPN circuit as being important for social affiliation, but that does not occur until later stages and not in young larvae, where most of the circuit diagram came from. Are there tools to easily validate these connections in juvenile fish that display the behavior of social affiliation to a conspecific and “fish-like” stimulus? To be clear, this is not a requirement of the authors – while it would strengthen their argument in these connections being important for natural behavior that the manuscript begins with, I recognize it may not be technologically feasible at this time.

Specific comments:

1. Line 31. What is the “fish-like” local acceleration? Does fish-like refer to the fact that the visual stimuli representing conspecifics are dots that they move more like a fish? Using this phrase in the abstract could be confusing to the reader who does not yet know what the stimulus is (if I’m interpreting this correctly). Using in the main text is fine.
2. Line 78 – A lot of this paragraph is dedicated to cortistatin, which seems randomly inserted. The link between this gene and the present study is unclear, especially since the neurons of interest in the DT that encode BM are not cort-positive.
3. There are lots of references to “custom scripts” in the methods. I suggest these scripts be uploaded to a public repository prior to publication to allow others to replicate the work.

Figures:

1. Figure 1: A would benefit from being larger – you could stack the bowls and make them larger. B – I suggest avoiding green/red if color is the only thing differentiating the groups. Do the colors in C mean anything?
2. Figure 2: A and B Would benefit labeling of bout-like (red?) and continuous (blue?) motion? For panel I, I suggest choosing different colors than red/green.

Congratulations on your beautiful work. - Lauren O’Connell

Referee #2 (Remarks to the Author):

The study identifies brain areas responding to biological (bout-like) motion in the zebrafish. A main finding is the detection of bout-motion responding (BPN) neurons in a nucleus of the dorsal thalamus (DT). BPN neurons in DT receive input from a subclass of tectal neurons (PVPN). Ablating tectal (PVPN and other) neurons affected behavior in the presence of conspecifics and altered responses of DT neurons. These results suggest that tectal PVPN neurons and the DT are part of a network that detects bout-like motion to control social affiliation.

The question how the brain detects biological motion is important and not well understood. The authors have made progress in this direction and the results will likely be of interest to many neuroscientists but several questions and expectations remain open. A main concern with this paper is that it unravels only a relatively small part of the network for visually driven social behavior. The DT is a likely candidate for a core component of the network and an excellent entry point. But the full picture is likely more complex (suggested for instance by the cfos data). A second concern is that the exact contributions of BPNs and PVPNs are not well defined because the functional

manipulations are not very specific. More information is desired to get a better grasp on the importance of the PVPN-DT connection for visually driven social behavior. It is possible that the DT and PVPNs are a central component of a visual-social circuit, but it is also possible that its role is less prominent. It would be terribly interesting to see results of more specific manipulations of PVPNs and of BPNs in DT. It would also be interesting to get insights into the mechanisms that generate sensitivity to bout motion. It has been a pleasure to read this paper because it addresses a question that, in my opinion, is very interesting and understudied, and I commend the authors for their achievements. Yet, further insights into the network and the specific function of the DT-PVPN nodes are desired.

Comments on the paper

A somewhat weak point of the paper is the limited specificity of the SAGFF(lf)81c driver for the loss-of-function experiment. Since it is expressed in many tectal neurons the observed phenotype cannot be linked specifically to PVPNs. More specific manipulations of these neurons should strengthen the conclusions. It would be very exciting to manipulate or ablate BPNs in DT directly to understand their function in social behavior. Less specific manipulations like bulk silencing/ablation of DT could also be informative. If it is difficult to find genetic drivers for these experiments it may be possible to address some of the questions by optogenetics.

The tectum contains neurons that are tuned to small moving objects and probably relevant for hunting and other behaviors. The observation that ablating a large fraction of tectal neurons affects behavioral responses to moving dots may then not come as a surprise. It would be desired to better understand the behavioral specificity of the ablations. The text mentions briefly that responses to looming stimuli appear to be intact. It would also be interesting to quantify the effect on other visual behaviors that depend on the tectum, particularly hunting (including eye convergence).

The identification of the “social behavior network” appears overstated and I feel that the description of the findings (c-fos) are not quite appropriate. The cfos response of at least some of the brain areas assigned to this network could be explained by the assumption that these areas are purely visual and not primarily associated with social stimuli/behavior. It is also not obvious why the “social behavior network” is defined as group 5 when responses in other groups (1 and 4) are more specific for social stimuli (conspecifics or conspecifics and bout-like) than the majority of the responses in group 5. The conclusions concerning the “social behavior network” are not convincing and the use of this term seems misleading to me.

The authors make the assumption that neurons involved in the detection of biological motion (bouts) should be tuned to parameters of natural bouts. I assume that swim bouts have not only a characteristic frequency but also length (distance). If so, it may be expected that neurons tuned to the retinal motion caused by conspecifics swimming in bouts should be tuned to specific combinations of speed, duration and distance (the combination varies with distance of the conspecific). It would be interesting to test whether this hypothesis is correct for BPNs in DT.

The paper describes that synaptic inputs to BPNs are coming from other DT neurons, PVPNs, and the

nucleus isthmi. But it is unclear what fraction of the synaptic inputs to BPNs come from these sources, and what the contribution of other sources is to the total input. The paper could be strengthened a lot by a more systematic and quantitative description of inputs to BPNs. Similarly, it should be interesting to quantify how much input PVPNs receive from RGCs in SFGS3/4 (fraction of all synapses, and/or fraction of RGC inputs).

Based on the cfos data it is obvious to ask whether PT and possibly mHr and Hi1 are involved in visually driven social behavior. Nevertheless, the paper focuses on DT. A rationale should be given for this focus (optical accessibility?). It would be very interesting to know how neurons in PT (and mHr/Hi1) are tuned, and whether these areas contain BPNs. These areas may be expected to contain BPNs because their responses to bouts are significant but responses to continuous motion are not.

In larva: is the expression of cort also not overlapping with BM selectivity at the single-neuron level? Fig S4d: Clusters of cort+ neurons and pth2+ neurons appear similar, suggesting that gene expression may overlap in individual neurons. The clusters appear to be only partially overlapping with the DT outline, suggesting expression may not be in BPNs. Correct?

More information about the tracing of neurons in EM data should be provided. For trans-synaptic tracing: how many synapses were identified as potential starting points? What fraction of these was chosen for trans-synaptic tracing? How were these synapses chosen, and can it be assumed that this choice was unbiased? What controls were performed to estimate the probability that branches may have been overlooked or incorrectly added to reconstructions?

Fig 4: it seems that BPNs in DT respond to a wide range of stimuli (Dots, gratings, loom). Average number of strongly responsive (top-scoring) cells seems to be similar for all stimuli. How can this observation be reconciled with the notion that this brain area is specialized to detect biological motion and the link to social behavior?

The text mentions that “One cluster in the PT stood out as selectively active with bout-like motion and real conspecifics...”. Fig 1e show a similar pattern of cfos changes for mHr and Hi1. Why are these areas not mentioned explicitly? Is there any difference in the cfos response between these areas and PT?

Fig 4c: using the regression score to assess neuronal responses is non-intuitive. It would be helpful to see also responses (DF/F) to each of the stimuli as examples plus analysis results pooled across neurons and fish (mean/median response intensity, mean/median number of responding neurons,...).

Minor comments:

The text says that “activation by real and virtual conspecifics overlapped in a subset of clusters including Hc1...” but Fig 1E shows that activity in Hc1 is unchanged or slightly reduced in bout-like and conspecific conditions. Typo?

How is a “premature” axonal projection defined? Axonal process with growth cone or axon without synapse?

How is an “axonal target” of a BPN defined? Is the definition based only on the projection of an axon to this brain region, or does it include synapses as a criterion? If synapses are not a criterion, this needs to be justified.

Referee #3 (Remarks to the Author):

This manuscript aims to reveal the neural circuits underlying the perception of visual cues related to social attraction. The authors leverage their previous findings that zebrafish respond to biological motion mimicking conspecifics swimming behavior with affiliation, also called shoaling. Using unbiased snapshot and volumetric two-photon microscopy, they query the brain for activity related to this specific type of motion, interpreted as socially activated. Among the 19 identified hotspots, including the caudal, rostral and intermediate hypothalamus, the dorsal thalamus (DT) stands out for containing a high concentration of neurons encoding kin-like motion acceleration, while being unresponsive to similar (global, continuous) motion which does not evoke behavioral, social affiliation. Next, they use electron microscopy to trace these putative DT bout preference neurons (BPNs), revealing their synaptic input from the optic tectum (TeO/ superior colliculus) and projections to ‘social network’ consisting of hypothalamic and other areas. Finally, chemogenetic ablation using nitroreductase in neurons of the TeO both disrupted DT activity to kin-like motion in larval zebrafish and reduced associated behavioral affiliation in 21 old juvenile fish. Together, they use the experimental power of the zebrafish to identify where social cues are encoded, presenting a neural pathway that can differentiate social kin-like motion from other forms of motion. This work presents new insights into how vertebrate brain activity is related to complex social behavior.

The significance of Kappel et al.’s research lies in examining the complex neural activity generated by motion cues that lead to a specific social response. They did this using an impressive multi-method approach that allowed them to first visualize snapshot brain activity after fixation and labeling of c-fos mRNA and then use volumetric two-photon microscopy to disentangle the encoding of kin-like visual input to other visual input, e.g., continuous motion. They harness the experimental power of the zebrafish to comprehensively describe all brain regions to reveal the functional topography, and intricately demonstrate how the zebrafish brain segregates into functional clusters, identifying neurons that specifically respond to visual input associated with biological motion that is interpreted as conspecific. After narrowing down these ‘special’ bout preference neurons (BPNs) with a high concentration in the dorsal thalamus, they leverage careful anatomical localization to find putative BPNs in electron microscopic stack to map their connectivity. Their study uses relatively standard technologies, but their combination of methods, approaches and questions are highly novel. Especially, the mapping of socially relevant cues in the brain is highly novel and informative in the context of the lab’s prior research on shoaling.

This study is very original in the sense that it describes functional imaging of juvenile (over 21 day old) transgenic zebrafish expressing fluorescent calcium indicators in almost all neurons to

holistically describe responsiveness to socially relevant, biological motion across the brain. This alone is an achievement that the field has been waiting for (previously imaging was either performed in larval fish or painstakingly in adults that require complex restraint). Therefore, one surprising aspect of this study is that it did not utilize this unique ability to a much greater extent and describe the methods in more detail. The fact that they can functionally identify neurons that may be part of the 'social recognition pathway' was a very interesting and provocative result, with the potential to open whole new lines of research. The inclusion of electron microscopic tracing is fantastic for mapping out the circuitry but lacks credibility as the link of the traced neurons and functionally identified BPNs is missing. Nonetheless, the ideas, methodologies, tools that were developed or used in this study represent a significant step to investigate how the zebrafish brain may process and respond to social cues.

Overall, the presented research appears highly rigorous, extremely well executed and the presented data are appropriately analyzed with relevant statistical methods, including an appropriate number of individuals. The figures are beautiful, easy to absorb, and the legends are clear and descriptive. The presented supplementary data answers many concerns and provides controls. And the materials and methods section are clear and detailed (with some exceptions, see below).

Despite the many strengths of the current manuscript, major concerns and some questions remain and require additional data, revisions, or discussion in the paper:

1) Conceptual concern #1: The presented data is largely correlational. While the identified DT-BPNs show strong and selective correlational activity, capable of differentiating between continuous motion and bout-like motion, the authors do not present evidence that they indeed mediate social behavior. Their ablation method of the TeO is not surprising given its importance in the visual processing of motion of moving objects. This is an excellent first-order experiment, but only coarsely maps its involvement. Furthermore, not all BPNs reside in the DT and the author's argument that the highest density of BPNs makes these neurons the likeliest or most important neurons is hard to believe.

a. Alternative experiments that would demonstrate specific BPN type involvement would be to selectively laser ablate BPNs in either the TeO or in the DT.

b. Optogenetic perturbations of these functionally defined BPNs could reveal their circuit role.

Evidently, one strength of their zebrafish preparation is that they can actually control (almost) all the neurons in a population of interest and look at the putative social network. Given the demonstrated high technical capability of this team, this should be within reach.

2) Conceptual concern #2: Kappel et al. describe the current stage-like development of social behaviors which emerges first as inter-individual repulsion in fish younger than 14 days and transitions into 'shoaling' (inter-individual attraction) beyond 14 days. Therefore, it is somewhat surprising that functional BPNs already exist in larval fish (less than 14 days). The authors argue that the functional maturation of these BPNs precedes shoaling and thus the transition from repulsion to attraction occurs downstream. However, if the existence of BPNs cannot explain the behavior, which part of the circuit is performing the critical computation? It is not surprising that these neurons possess specific receptive fields for bout-like motion to exist as these fish grew up for 21 days in the company of other (20-25 individuals are mentioned) zebrafish swimming in bout-like patterns as

their main visual input.

At a minimum, the authors should consider altering visual input over development by rearing in social, that is visual, isolation from the 'bout like' visual stimulus represented by other fish. In addition, changing the visual environment for example with dark rearing or in strobe lighting conditions would resolve whether these neurons are a result of rearing conditions.

While Larsch et al. 2018 show that rearing fish in isolation does not affect overall attraction or optimal bout interval selectivity, it would still be an important test to show that BPNs are unchanged. If that is the case, BPNs appear to possess innate specific tuning that possibly could be species-specific as different larval and juvenile fish show different swimming patterns (e.g., *Danio rerio* larva do not swim in bouts but rather continuously even as larva).

Minor: please provide context on whether or how the social cue encoding by BPNs is universal/ conserved. Discuss whether this may be a specific case of visual social cue encoding.

3) The explanation that the critical computation occurs downstream without further explanation is somewhat disappointing. If the claim of the study is the discovery of a tectal thalamic pathway that drives a 'social network' they failed in the sense that no direct evidence links BPNs to this social network. Understandably, using EM on functionally identified BPNs is difficult. Yet, any method that would establish a link between BPNs and any downstream partner would be highly desirable to make this statement. For example, imaging of the hypothalamus while stimulating with social/ control stimuli may provide convincing evidence.

4) The authors could have done more to explain how they achieved imaging in juvenile zebrafish, a method that has long been thought elusive and too difficult. Is there any behavioral indication that the social behavioral patterns are still intact in the configuration under the microscope? How long is the preparation stable? Are there any differences in handling, recorded variables that indicate health? Why was no imaging performed in the 'social network'? Please elaborate in the methods section exactly how and provide supplementary figures presenting the configuration.

5) Please provide a supplementary movie/ imaging stack (data) of the functional imaging in juvenile fish.

6) It would help if the authors discussed how the BPN selectivity is created and how their responses may serve to separate behaviors. What is the relationship between TeO and DT BPNs?

7) It would be great if they had expanded on the broader view in the literature on how social signals are encoded. As is, the reader learns almost nothing about how these results can be interpreted in the larger field of social cue processing.

8) Another limitation is that the authors focused only on the BPNs without probing or discussing the relevance of the downstream processing and how the social attraction could be implemented. It

would be better if this were discussed in more detail in both the main text and the discussion. And in the future, it would be great to see if all neurons with shared anatomical characteristics with BPNs process social information. This team recently built a fantastic technique described in Kramer et al. 2019- Function-guided inducible morphological analysis (FuGIMA). This could be used to elucidate whether actual, not putative BPN projection patterns and compared to the EM data.

9) Given the presented data in Figure 2, why was there no imaging performed in tectal ablated 21-day old fish (only behavior, Figure 4)?

10) Finally, a cell-type-specific model that would illustrate the predictions for perturbation experiments would be much appreciated. Currently, the understanding of the circuit computation appears to be limited to the mere existence of BPNs in the TeO or DT. Understanding what functional role they play likely requires a model or clear hypothesis of how their activity leads to social attraction or repulsion.

If these points would be addressed, new data demonstrate causality in this circuit including BPNs, and whether/how this circuit can lead to repulsion in larva and attraction in juveniles, the presented research could reach a very high impact level, justifying the publication in this journal. These insights into how vertebrate brains can use visual signals as social cues to direct behavior are very important for our understanding of complex neural processing.

Author Rebuttals to Initial Comments:

Point by point reply to reviewers comments

Referee #1 (Remarks to the Author):

In this manuscript, the authors provide evidence for a tectal-thalamic circuit that is important for shoaling/social affiliation in zebrafish. While many genes/neuromodulators have been associated with shoaling/social affiliation in teleosts, a neural circuit for these behaviors is not yet characterized, making this paper an exciting step forward. The authors use a number of different approaches to tackle this questions, including in vivo imaging, neural activity mapping, chemogenetics, and utilizing an electron microscopy-derived connectome map of the larval zebrafish brain. The data is robust, although in its current form, it is sometime unclear why the authors made certain choices in which brain regions/cells to investigate.

General comments:

(1)

1. Social behavior network: I appreciate that the authors are setting their results within the wider literature of brain networks that govern social behavior across vertebrates – the Social Behavior Network originally proposed by Newman and composed of the medial amygdala, lateral septum, preoptic area, anterior hypothalamus, ventromedial hypothalamus, and paraventricular/central gray. However, I think that the application of this well-established network referencing very specific brain regions in vertebrates to non-analogous (?) brain regions in teleosts that are more active in real/virtual conspecifics is misleading. For example, line 96 states that the authors have identified a social behavior network whose activity is modulated by real and virtual conspecifics. However, based on my interpretation of Figure 1e, this could include PT, mHr, and Hi1, as the other brain regions are also activated by continuous (non-fish-like) motion. The authors should consider stating that brain regions PT, mHr, and Hi1 are analogous to in other brains regions in the social behavior network, considering adopting a different name, or propose expanding the current network to include these regions in the context of shoaling. Similarly, the PVPN -> DT-BPN -> POA pathway seems to be the only connection to the classical “Social Behavior Network”, unless I am misinterpreting fish brain regions. I do not dispute that these DT neurons project to places important for social behavior but caution the use of this phrase as something that may be confusing to readers familiar with established literature on the topic.

We have toned down the parts mentioning the social behavior network (SBN) for a more nuanced description of likely homologies. We now specifically discuss that the teleost preoptic area and rostral hypothalamus are likely homologous with paraventricular, preoptic and other anterior hypothalamic areas in other vertebrates.

(2)

2. Looking at the data in 1e, group 5, it is unclear to me why the authors decided to focus on the dorsal thalamus and not the other brain regions that were activated in bout-like and conspecific situations, but not with the continuous stimuli.

The reviewer is correct, DT is not the region showing strongest selectivity of *c-fos* signal for bout-like and conspecific stimuli versus continuous stimuli. However, we ended up focusing on DT because it was the area with the highest density of stimulus-driven BPNs of those areas covered by volumetric imaging. PT, mHr and Hi1 are interesting candidates for future studies that will require modifications to the imaging preparation for better signal-to-noise ratio at greater depth.

Also see related comment (15) by reviewer 2.

We have added a statement to emphasize our focus on visual detection.

(3)

3. It is interesting that there is activation of tectal neurons only with real conspecifics (Fig 1e) and not with the other visual stimuli. Do the authors have ideas for what this means for visual processing of real versus virtual stimuli? The chemoablations of these neurons do influence the affiliative behavior, so perhaps there is only a tiny subset that encodes conspecific acceleration/speed?

We wonder if there was a misunderstanding with figure 1e. We would like to clarify that cells in the ventral tectum (TeOv) do show *c-fos* activation with virtual stimuli (did the reviewer miss this?) while anterior and dorsal tectum (TeOa, TeOd) show activation with real conspecifics (as the reviewer states above). We think the most parsimonious explanation for this difference is the stimulus location in the visual field: Virtual stimuli are shown from below (ventral), real conspecifics typically swim in the same horizontal plane. The dorso-ventral retinotopy of the tectum can therefore explain differential activation. In addition, real conspecifics provide more stimulus features than our virtual stimuli, including color, shape and depth. While apparently not necessary for shoaling, these features of real conspecifics may drive additional tectal activity.

We added clarifying statements to the text.

(4)

4. The authors propose a PVPN -> DT-BPN circuit as being important for social affiliation, but that does not occur until later stages and not in young larvae, where most of the circuit diagram came from. Are there tools to easily validate these connections in juvenile fish that display the behavior of social affiliation to a conspecific and “fish-like” stimulus? To be clear, this is not a requirement of the authors – while it would strengthen their argument in these connections being important for natural behavior that the manuscript begins with, I recognize it may not be technologically feasible at this time.

To demonstrate necessity of the tectum-DT circuit for shoaling in juveniles, we have now repeated the tectal (SAGFF(lf)81c) ablation with subsequent DT imaging in juveniles and

added DT chemogenetic and laser ablations in juveniles. These experiments more directly implicate the tectum-DT circuit in shoaling. We do not have genetic tools to label specifically or stochastically the PVPN->DT-BPN connections in juveniles but we think it is most parsimonious to assume they persist from larval to juvenile stage.

Specific comments:

(5)

1. Line 31. What is the “fish-like” local acceleration? Does fish-like refer to the fact that the visual stimuli representing conspecifics are dots that they move more like a fish? Using this phrase in the abstract could be confusing to the reader who does not yet know what the stimulus is (if I’m interpreting this correctly). Using in the main text is fine.

This is now clarified.

(6)

2. Line 78 – A lot of this paragraph is dedicated to cortistatin, which seems randomly inserted. The link between this gene and the present study is unclear, especially since the neurons of interest in the DT that encode BM are not cort-positive.

Cortistatin is a genetic landmark for the area of the DT-BPN cluster. As such, our finding of overlap of *cort* with the *c-fos* signal at the regional level but not at the cellular level seems worth reporting.

However, we agree that *cort* doesn’t add substance to the main narrative so we shortened this part in the main text but kept the supplementary figure.

(7)

3. There are lots of references to “custom scripts” in the methods. I suggest these scripts be uploaded to a public repository prior to publication to allow others to replicate the work.

Thank you for this suggestion. We are uploading our entire analysis workflow and experimental control scripts to a public bitbucket repository (<https://bitbucket.org/mpinbaierlab/kappeletal2022>).

This includes the following ‘custom scripts’

- Bonsai workflows for behavior experiments with virtual and real conspecifics.
- Python notebooks for quantifying behavior.
- Python notebooks for 2p imaging analysis.

(8)

1. Figure 1: A would benefit from being larger – you could stack the bowls and make them larger. B – I suggest avoiding green/red if color is the only thing differentiating the groups. Do the colors in C mean anything?

Thank you for these suggestions - which we have implemented. The colors did not have meaning, except aiding visualization of many separate clusters.

(9)

2. Figure 2: A and B Would benefit labeling of bout-like (red?) and continuous (blue?) motion? For panel I, I suggest choosing different colors than red/green.

Thank you for these suggestions - which we have implemented.

Referee #2 (Remarks to the Author):

The study identifies brain areas responding to biological (bout-like) motion in the zebrafish. A main finding is the detection of bout-motion responding (BPN) neurons in a nucleus of the dorsal thalamus (DT). BPN neurons in DT receive input from a subclass of tectal neurons (PVPN). Ablating tectal (PVPN and other) neurons affected behavior in the presence of conspecifics and altered responses of DT neurons. These results suggest that tectal PVPN neurons and the DT are part of a network that detects bout-like motion to control social affiliation.

The question how the brain detects biological motion is important and not well understood. The authors have made progress in this direction and the results will likely be of interest to many neuroscientists but several questions and expectations remain open. A main concern with this paper is that it unravels only a relatively small part of the network for visually driven social behavior. The DT is a likely candidate for a core component of the network and an excellent entry point. But the full picture is likely more complex (suggested for instance by the cfos data). A second concern is that the exact contributions of BPNs and PVPNs are not well defined because the functional manipulations are not very specific. More information is desired to get a better grasp on the importance of the PVPN-DT connection for visually driven social behavior. It is possible that the DT and PVPNs are a central component of a visual-social circuit, but it is also possible that its role is less prominent. It would be terribly interesting to see results of more specific manipulations of PVPNs and of BPNs in DT. It would also be interesting to get insights into the mechanisms that generate sensitivity to bout motion. It has been a pleasure to read this paper because it addresses a question that, in my opinion, is very interesting and understudied, and I commend the authors for their achievements. Yet, further insights into the network and the specific function of the DT-PVPN nodes are desired.

Comments on the paper

(10)

A somewhat weak point of the paper is the limited specificity of the SAGFF(lf)81c driver for the loss-of-function experiment. Since it is expressed in many tectal neurons the observed phenotype cannot be linked specifically to PVPNs. More specific manipulations of these neurons should strengthen the conclusions. It would be very exciting to manipulate or ablate BPNs in DT directly to understand their function in social behavior. Less specific manipulations like bulk silencing/ablation of DT could also be informative. If it is difficult to find genetic drivers for these experiments it may be possible to address some of the questions by optogenetics.

We agree that a demonstration of the necessity of DT for shoaling was missing from the paper. We now show new data from two experiments directly testing the necessity of DT for shoaling in juveniles:

(1) We generated a new transgenic line to express a neuronally restricted nitroreductase in DT using the s1026tEt Gal4 driver line whose expression is strongest in DT and parts of (dorsal) VT and (ventral) pretectum, with additional expression in habenula and ventral telencephalon.

(2) We performed bilateral laser ablations in the area of the DT-BPN nucleus.

Both manipulations strongly reduced shoaling while leaving control behaviors intact.

(11)

The tectum contains neurons that are tuned to small moving objects and probably relevant for hunting and other behaviors. The observation that ablating a large fraction of tectal neurons affects behavioral responses to moving dots may then not come as a surprise. It would be desired to better understand the behavioral specificity of the ablations. The text mentions briefly that responses to looming stimuli appear to be intact. It would also be interesting to quantify the effect on other visual behaviors that depend on the tectum, particularly hunting (including eye convergence).

We agree that the SAGFF(lf)81c line is not specific to PVPNs and its ablation likely affects other tectum dependent behaviors. We also agree that specifically the loss in attraction upon SAGFF(lf)81c ablation is not surprising. However, we believe that the SAGFF(lf)81c is unexpectedly informative for demonstrating that social repulsion (and loom responses) don't depend on these neurons, and, thus, attraction and repulsion have different neuronal requirements. The SAGFF(lf)81c ablation is also useful for determining that DT is downstream of the ablated neurons via functional 2p calcium imaging. We do not want to make the point that SAGFF(lf)81c ablation only affects shoaling. In fact, we strongly expect hunting to be defective in this line too, because hunting requires the tectum (e.g. recently revisited in Foerster et al. 2021).

With the new DT ablations, the demonstration of necessity of the BPN circuit no longer hinges on the SAGFF(lf)81c ablation and we have toned down the interpretation of SAGFF(lf)81c results in the text.

(12)

The identification of the “social behavior network” appears overstated and I feel that the description of the findings (c-fos) are not quite appropriate. The cfos response of at least some of the brain areas assigned to this network could be explained by the assumption that these areas are purely visual and not primarily associated with social stimuli/behavior. It is also not obvious why the “social behavior network” is defined as group 5 when responses in other groups (1 and 4) are more specific for social stimuli (conspecifics or conspecifics and bout-like) than the majority of the responses in group 5. The conclusions concerning the “social behavior network” are not convincing and the use of this term seems misleading to me.

Thank you for pointing out these overstatements. As described in response to comment 1 (reviewer 1), we have now toned down our statements regarding the SBN.

(13)

The authors make the assumption that neurons involved in the detection of biological motion (bouts) should be tuned to parameters of natural bouts. I assume that swim bouts have not only a characteristic frequency but also length (distance). If so, it may be expected that neurons tuned to the retinal motion caused by conspecifics swimming in bouts should be tuned to specific combinations of speed, duration and distance (the combination varies with distance of the conspecific). It would be interesting to test whether this hypothesis is correct for BPNs in DT.

Figures 2G&H already provide evidence that naturalistic combinations of speed and frequency of swim bouts are most effective in driving DT-BPNs, and figure 2I shows that these neurons also respond strongly to naturalistic trajectories (Dot shoaling condition). As we understand the question, the reviewer asks if stimulus distance modulates the preferred speed. The reasoning being that a dot moving at a specific (allocentric) absolute speed would generate a smaller change in (egocentric) visual angle when it is far away vs. nearby (whereas bout frequency does not change with distance). For our experiments, we have positioned a 4 mm dot at 20 mm distance to the fish, corresponding to the preferred size and distance in free-swimming conditions (Larsch & Baier 2018). The experiment suggested by the reviewer would address how robust is the apparent speed tuning to variable stimulus distance.

While this is an interesting psychophysical extension of our DT-BPN characterization, we believe this point is not central enough to justify the considerable effort necessary to do this properly. In an embedded preparation, the perception of stimulus distance is, to a first approximation, ambiguous with stimulus size. See paragraph 3.5 in (Orger & Polavieja: <https://www.annualreviews.org/doi/full/10.1146/annurev-neuro-071714-033857>) for a discussion of this ambiguity. In addition, due to increasingly shallow viewing angles, more distant stimuli will become increasingly distorted on the flat projection surface underneath the fish in our setup. As a result, we would not be able to clearly attribute changes in DT-BPN responses to distance, vs. changes in size/distortion/luminance. Overcoming this challenge would require re-designing the stimulus arena, for example using curved displays. But even then, it is unclear if immobilized fish can properly perceive distance (see citation above). Therefore, we hope the reviewer will agree that this experiment is beyond the scope of this manuscript.

(14a)

The paper describes that synaptic inputs to BPNs are coming from other DT neurons, PVPNs, and the nucleus isthmi. But it is unclear what fraction of the synaptic inputs to BPNs come from these sources, and what the contribution of other sources is to the total input. The paper could be strengthened a lot by a more systematic and quantitative description of inputs to BPNs.

As suggested by the reviewer, we systematically quantified the fraction of synaptic inputs to BPNs. We randomly selected 10 input synapses onto 3 putative BPNs each (30 synapses in total), and identified the input partner cells. On average, 26.7% of all inputs were coming from PVPNs, and similarly 26.7% from other DT-pBPNs. 16.7% of inputs arrived from the ventral thalamus, 10% from the nucleus isthmi, 6.7% each from the superior ventral medulla oblongata and from the torus semicircularis, and 3.3% of inputs each from the hypothalamus and cerebellum. Overall, the majority of synaptic inputs (76.7%) came from ipsilateral neurons. This additional tracing effort more than doubled the number of identified tectal projection neurons, which are presynaptic to pBPNs. We added this quantitative description to the results (page 10) and methods ('Electron microscopy and segmentation of mapzebrain regions'). Figure 3 was updated accordingly.

(14b)

Similarly, it should be interesting to quantify how much input PVPNs receive from RGCs in SFGS3/4 (fraction of all synapses, and/or fraction of RGC inputs).

Analogous to the procedure described above, we also quantified the fraction of synaptic inputs to PVPNs in the SFGS3/4 layer. We again randomly selected 10 input synapses onto 3 PVPNs each (30 synapses in total) and identified the input partners. On average, 55% of inputs came from RGC axons, 35% from tectal interneurons (PVINs), and interestingly, 10% came from other PVPNs, some of which also made downstream connections with BPNs. We added these new results on page 10.

(15)

Based on the cfos data it is obvious to ask whether PT and possibly mHr and Hi1 are involved in visually driven social behavior. Nevertheless, the paper focuses on DT. A rationale should be given for this focus (optical accessibility?). It would be very interesting to know how neurons in PT (and mHr/Hi1) are tuned, and whether these areas contain BPNs. These areas may be expected to contain BPNs because their responses to bouts are significant but responses to continuous motion are not.

We have now clarified our focus on sensory areas in the text.

We agree that PT and mHR and Hi1 are excellent candidates for further investigation. As discussed in response to comment (2) by Reviewer 1, our study primarily examines visual detection of social signals as an entry point for further investigation of downstream processing. We hope the reviewers agree that in this context the focus on tectum and thalamus is justified. In addition, optical access is indeed an issue for imaging hypothalamic areas in the juvenile that will likely require creative preparations for imaging fish upside down, from the ventral side, via adaptive optics, or 3-photon imaging. While BPNs in these regions are a plausible expectation, we also predict that it will become increasingly important to ensure intact behavioral readouts from the imaging preparation for interpreting the presence or absence of such signals. In the case of larval zebrafish hunting for

example, the activity of certain pretectal neurons reflects a combination of prey detection and hunting decision (Antinucci et al. <https://elifesciences.org/articles/48114>).

It will be fascinating to determine how neuronal activation by social cues becomes increasingly uncoupled from sensory stimulation and integrated with the internal state in hypothalamic brain areas. However, we believe this is beyond the scope of our manuscript.

(16)

In larva: is in the expression of cort also not overlapping with BM selectivity at the single-neuron level? Fig S4d: Clusters of cort+ neurons and pth2+ neurons appear similar, suggesting that gene expression may overlap in individual neurons. The clusters appear to be only partially overlapping with the DT outline, suggesting expression may not be in BPNs. Correct?

Our manuscript currently does not address developmental changes in (co-)expression of the marker genes *cort* and *pth2* in BM selective neurons (BPNs).

As reviewer 1 pointed out, we agree that our detour into *cort* expression in the main text is somewhat disconnected from the main message. We think it remains valuable to report *cort* and *pth2* as landmarks for the location of the BPN cluster, even if these markers do not label BPNs themselves. This is particularly interesting in light of the recent discovery that *pth2* expression in this area is regulated by the density of conspecifics detected via mechanosensation (Anneser et al. 2020) and implies the possibility that multimodal social cues are integrated in this area. How these multisensory cues are integrated will be a fascinating future avenue.

There is an ongoing molecular characterization of thalamic markers in a separate project in our lab and we see conceptual issues with shoaling-associated activity mapping in larvae, as described below. Therefore we believe that further characterization of *cort* and *pth2* relative to BM selectivity specifically in larvae would not substantially add to the current manuscript.

We have now focused the description of *cort* in the main text on its use as a landmark.

Below, we present additional data regarding larval co-expression of *cort* and *pth2* - as requested by reviewer 2, and a description of the conceptual issues of larval BM activity mapping. However, we suggest that these data not be included in the paper.

Our new HCR data shows that larval *cort+* and *pth2+* clusters overlap and, indeed, there is some overlap in gene expression also at the individual neuron level (see arrowheads in the figure below).

In a recent study from our group (Sherman et al, preprint at <https://www.biorxiv.org/content/10.1101/2021.11.15.468630v1>) we show that *cort* labels a subset of *pth2*+ neurons in the thalamus Fig 2b and S1&S2 in Sherman et al.)

The reviewer asked specifically in larvae about overlap of the *cort* and *pth2* markers with BM selectivity. However, using our current tools, we cannot directly answer this question in larvae. In juveniles, we examined (in the same animals) *cort* expression in neurons with high *cfos* induction after shoaling (Figure S2). In contrast, for larvae we show *pth2* and *cort* expression co-registered with a BPN region derived from larval GCaMP imaging in a separate experiment in a different set of animals.

Since larvae don't shoal with visual BM stimuli, we do not expect substantial *cfos* induction with such stimuli from free-swimming larvae. This expectation was confirmed in a pilot experiment. We interpret this preliminary result as an indication that larval BPN responses detected in immobilized non-social larvae via GCaMP imaging are the result of 'forcing' animals to observe the visual stimulus. In contrast, free-swimming larvae that are behaviorally unresponsive to such stimuli are, on average, too far away from the stimulus for their brains to be activated by it to levels reflected in *cfos*. Thus, a thorough analysis of differences and similarities of *cfos* induction between free-swimming and immobilized

animals would be required to address this point properly. We expect the likely outcome is that *pth2* labels a small subset of DT-BPNs (there are many more BPNs than *pth2*⁺ neurons in larvae).

(17)

More information about the tracing of neurons in EM data should be provided. For trans-synaptic tracing: how many synapses were identified as potential starting points? What fraction of these was chosen for trans-synaptic tracing? How were these synapses chosen, and can it be assumed that this choice was unbiased? What controls were performed to estimate the probability that branches may have been overlooked or incorrectly added to reconstructions?

For the initial submission of the manuscript, we randomly picked 34 putative BPNs and for each cell we randomly selected 1-3 synapses for trans-synaptic tracing. As mentioned above, we now added a systematic quantification of BPN partners by randomly selecting at least 10 synapses per BPN. We assume that this random selection was unbiased. For all of these synapses, we identified the partner cells.

Proofreading of our automatic segmentation mainly involves the correction of false splits (we quantified on average about 100 human merge interactions per cell). Thus, it is possible that additional branches (mainly thin axons) may have been overlooked. However, when branches were added to a neurite by a human expert, they certainly have been part of the same cell, thus incorrect additions are basically excluded.

(18)

Fig 4: it seems that BPNs in DT respond to a wide range of stimuli (Dots, gratings, loom). Average number of strongly responsive (top-scoring) cells seems to be similar for all stimuli. How can this observation be reconciled with the notion that this brain area is specialized to detect biological motion and the link to social behavior?

The reviewer had the impression that DT-BPNs responded to our grating and loom control stimuli. We believe this is a misunderstanding of our Figure 4c and we apologize for the unclear visualization of the data. **Each column in figure 4c of our initial submission showed a different set of neurons** that respond strongest to the respective stimulus (top 10% for each stimulus). Only the rightmost bar plot specifically quantified the number of BPNs in DT. The plot was intended to show an effect of the ablation on responses to different stimuli and the location of responding neurons.

Figure 4 is now thoroughly revised and the data mentioned by the reviewer is now re-analyzed as part of supplementary figure S8. Instead of regression scores, we now use mean dF/F scores as in figure 2. Further, we restrict our comparison of ablated and control animals specifically on anatomical areas: dorsal thalamus for BPNs (fig. 4e-h) and responses to other dot stimuli, and pretectum for grating and looming stimuli. We have added a control panel for the dataset from figure 2b-g showing that DT-BPNs do not respond strongly to translational grating or looming stimuli.

(19)

The text mentions that “One cluster in the PT stood out as selectively active with bout-like motion and real conspecifics...”. Fig 1e show a similar pattern of cfos changes for mHr and Hi1. Why are these areas not mentioned explicitly? Is there any difference in the cfos response between these areas and PT?

The PT signal visually stood out from the raw fluorescence data because it has high absolute intensity whereas mHr and Hi1 signals are of overall lower intensity (Figure S2) (but similar differences between groups in terms of effect size).

(20)

Fig 4c: using the regression score to assess neuronal responses is non-intuitive. It would be helpful to see also responses (DF/F) to each of the stimuli as examples plus analysis results pooled across neurons and fish (mean/median response intensity, mean/median number of responding neurons, ...).

Thank you for this feedback. We have changed the analysis to dF/F. Figure 4f now shows dF/F time series from individual neurons and group means. In addition, each light datapoint in Fig. S8c&d represents the mean responses across neurons for one animal, and dark data points represent the group mean across animals.

Minor comments:

(21)

The text says that “activation by real and virtual conspecifics overlapped in a subset of clusters including Hc1...” but Fig 1E shows that activity in Hc1 is unchanged or slightly reduced in bout-like and conspecific conditions. Typo?

Thank you for catching this typo!

(22)

How is a “premature” axonal projection defined? Axonal process with growth cone or axon without synapse?

Premature axons were defined by growth cones (see Methods, ‘Electron microscopy and segmentation of mapzebrain regions’).

(23)

How is an “axonal target” of a BPN defined? Is the definition based only on the projection of an axon to this brain region, or does it include synapses as a criterion? If synapses are not a criterion, this needs to be justified.

All BPNs we investigated, which project axons to other brain regions, also made synaptic contacts in these regions. We added this explanation on page 10.

Referee #3 (Remarks to the Author):

This manuscript aims to reveal the neural circuits underlying the perception of visual cues related to social attraction. The authors leverage their previous findings that zebrafish respond to biological motion mimicking conspecifics swimming behavior with affiliation, also called shoaling. Using unbiased snapshot and volumetric two-photon microscopy, they query the brain for activity related to this specific type of motion, interpreted as socially activated. Among the 19 identified hotspots, including the caudal, rostral and intermediate hypothalamus, the dorsal thalamus (DT) stands out for containing a high concentration of neurons encoding kin-like motion acceleration, while being unresponsive to similar (global, continuous) motion which does not evoke behavioral, social affiliation. Next, they use electron microscopy to trace these putative DT bout preference neurons (BPNs), revealing their synaptic input from the optic tectum (TeO/ superior colliculus) and projections to 'social network' consisting of hypothalamic and other areas. Finally, chemogenetic ablation using nitroreductase in neurons of the TeO both disrupted DT activity to kin-like motion in larval zebrafish and reduced associated behavioral affiliation in 21 old juvenile fish. Together, they use the experimental power of the zebrafish to identify where social cues are encoded, presenting a neural pathway that can differentiate social kin-like motion from other forms of motion. This work presents new insights into how vertebrate brain activity is related to complex social behavior.

The significance of Kappel et al.'s research lies in examining the complex neural activity generated by motion cues that lead to a specific social response. They did this using an impressive multi-method approach that allowed them to first visualize snapshot brain activity after fixation and labeling of c-fos mRNA and then use volumetric two-photon microscopy to disentangle the encoding of kin-like visual input to other visual input, e.g., continuous motion. They harness the experimental power of the zebrafish to comprehensively describe all brain regions to reveal the functional topography, and intricately demonstrate how the zebrafish brain segregates into functional clusters, identifying neurons that specifically respond to visual input associated with biological motion that is interpreted as conspecific. After narrowing down these 'special' bout preference neurons (BPNs) with a high concentration in the dorsal thalamus, they leverage careful anatomical localization to find putative BPNs in electron microscopic stack to map their connectivity. Their study uses relatively standard technologies, but their combination of methods, approaches and questions are highly novel. Especially, the mapping of socially relevant cues in the brain is highly novel and informative in the context of the lab's prior research on shoaling.

This study is very original in the sense that it describes functional imaging of juvenile (over 21 day old) transgenic zebrafish expressing fluorescent calcium indicators in almost all neurons to holistically describe responsiveness to socially relevant, biological motion across the brain. This alone is an achievement that the field has been waiting for (previously imaging was either performed in larval fish or painstakingly in adults that require complex restraint). Therefore, one surprising aspect of this study is that it did not utilize this unique ability to a much greater extent and describe the methods in more detail. The fact that they

can functionally identify neurons that may be part of the ‘social recognition pathway’ was a very interesting and provocative result, with the potential to open whole new lines of research. The inclusion of electron microscopic tracing is fantastic for mapping out the circuitry but lacks credibility as the link of the traced neurons and functionally identified BPNs is missing. Nonetheless, the ideas, methodologies, tools that were developed or used in this study represent a significant step to investigate how the zebrafish brain may process and respond to social cues.

Overall, the presented research appears highly rigorous, extremely well executed and the presented data are appropriately analyzed with relevant statistical methods, including an appropriate number of individuals. The figures are beautiful, easy to absorb, and the legends are clear and descriptive. The presented supplementary data answers many concerns and provides controls. And the materials and methods section are clear and detailed (with some exceptions, see below).

Despite the many strengths of the current manuscript, major concerns and some questions remain and require additional data, revisions, or discussion in the paper:

We appreciate the very detailed and nuanced feedback on our manuscript. In summary, reviewer 3 recognizes our discovery of a circuit that detects biological motion (BM) as “highly novel and informative” and yielding “very interesting and provocative result, with the potential to open whole new lines of research”. However, the reviewer, in several comments, suggests substantial additional experiments regarding the ontogeny of BM detection, its processing by downstream circuits and how this relates to the behavioral switch from the non-social larval stage to the social juvenile stage. Some of these points are suggested as “future experiments” (comment 8). We agree with the reviewer that these are highly relevant and fascinating topics for future research and we have implemented many of their suggestions. At the same time, we argue that explaining the onset of social affiliation at the level of BM processing in circuits downstream of DT is well beyond the scope of this manuscript. In addition, suggestions towards a functional dissection of hypothalamic circuits with thalamic stimulation are technically not feasible in juveniles using current imaging methods and transgenes. As we specify in the title and abstract of our manuscript, we think our primary discovery is that a tecto-thalamic circuit detects BM and is required for juvenile shoaling, and a synapse-level anatomical characterization of its connectivity in the context of snapshot activity maps. Identification of a visual social input pathway is the truly innovative insight that remains elusive in many other social behavior paradigms and model systems. Future work will delineate the downstream circuits and ontogeny.

We have modified the summary statement in the abstract “Together, we discovered a tecto-thalamic pathway that drives ~~a core network for~~ social affiliation” to reflect the scope of our experiments and interpretations more clearly.

Below follows our point-by-point response to your specific comments:

(24)

1) Conceptual concern #1: The presented data is largely correlational. While the identified DT-BPNs show strong and selective correlational activity, capable of differentiating between continuous motion and bout-like motion, the authors do not present evidence that they indeed mediate social behavior. Their ablation method of the TeO is not surprising given its importance in the visual processing of motion of moving objects. This is an excellent first-order experiment, but only coarsely maps its involvement. Furthermore, not all BPNs reside in the DT and the author's argument that the highest density of BPNs makes these neurons the likeliest or most important neurons is hard to believe.

a. Alternative experiments that would demonstrate specific BPN type involvement would be to selectively laser ablate BPNs in either the TeO or in the DT.

b. Optogenetic perturbations of these functionally defined BPNs could reveal their circuit role. Evidently, one strength of their zebrafish preparation is that they can actually control (almost) all the neurons in a population of interest and look at the putative social network. Given the demonstrated high technical capability of this team, this should be within reach.

We appreciate the concern of weak causality, in particular with respect to DT neurons.

We have addressed this concern with two new tests of necessity via ablation (Figure 4): First, we generated a new transgenic zebrafish line for chemogenetic ablation of DT. Second, we established a DT laser ablation protocol. These experiments strongly suggest a specific role of DT in social affiliation and, to our knowledge, represent the first example of in-depth functional circuit dissection in juvenile zebrafish.

Optogenetic perturbation specifically of DT BPNs via holography at a depth of 250 um in combination with hypothalamic imaging at 400-500 um depth is currently not feasible given the required laser power and a lack of BPN-specific genetic driver lines.

(25)

2) Conceptual concern #2: Kappel et al. describe the current stage-like development of social behaviors which emerges first as inter-individual repulsion in fish younger than 14 days and transitions into 'shoaling' (inter-individual attraction) beyond 14 days. Therefore, it is somewhat surprising that functional BPNs already exist in larval fish (less than 14 days). The authors argue that the functional maturation of these BPNs precedes shoaling and thus the transition from repulsion to attraction occurs downstream. However, if the existence of BPNs cannot explain the behavior, which part of the circuit is performing the critical computation? It is not surprising that these neurons possess specific receptive fields for bout-like motion ~~to~~ exist as these fish grew up for 21 days in the company of other (20-25 individuals are mentioned) zebrafish swimming in bout-like patterns as their main visual input.

At a minimum, the authors should consider altering visual input over development by rearing in social, that is visual, isolation from the 'bout like' visual stimulus represented by other fish. In addition, changing the visual environment for example with dark rearing or in strobe lighting conditions would resolve whether these neurons are a result of rearing conditions.

While Larsch et al. 2018 show that rearing fish in isolation does not affect overall attraction or optimal bout interval selectivity, it would still be an important test to show that BPNs are unchanged. If that is the case, BPNs appear to possess innate specific tuning that possibly could be species-specific as different larval and juvenile fish show different swimming patterns (e.g., Danionella translucida larva do not swim in bouts but rather continuously even as larva).

The reviewer raises the possibility that BPNs might develop their receptive fields for bout-like motion patterns by observing conspecifics and suggests a social isolation experiment. Since we already observe BPNs at 6 dpf, the relevant experience dependent maturation should have occurred by then. We have now experimentally tested this prediction and show in an additional figure panel (S5g) that larval BPNs also exist after full social isolation until 6 dpf. This shows that BPN tuning is not a mere epiphenomenon of observing conspecifics. Instead, it is the result of an innate developmental process, consistent with innate behavioral responses to conspecifics. We think this finding, in combination with our new DT ablation, provides strong evidence for a parsimonious model: Sensory tuning to BM in DT develops innately at the larval stage; it is necessary for juvenile shoaling which also requires maturation in other brain areas. We have changed our previously more specific hypothesis of changes ‘downstream’ accordingly.

(26)

Minor: please provide context on whether or how the social cue encoding by BPNs is universal/ conserved. Discuss whether this may be a specific case of visual social cue encoding.

Motion cues emerge as critical features of social cues across species. Detection of self-like swim bouts may be a (fish-)specific case. More generally, encoding of conspecific motion in homologous brain areas dedicated to social information may be conserved. Due to the word limit, we had only included a brief statement in the introduction and now added one additional sentence in the discussion with references to publications in this field.

(27)

3) The explanation that the critical computation occurs downstream without further explanation is somewhat disappointing. If the claim of the study is the discovery of a tectal thalamic pathway that drives a ‘social network’ they failed in the sense that no direct evidence links BPNs to this social network. Understandably, using EM on functionally identified BPNs is difficult. Yet, any method that would establish a link between BPNs and any downstream partner would be highly desirable to make this statement. For example, imaging of the hypothalamus while stimulating with social/ control stimuli may provide convincing evidence.

The reviewer correctly states that we do not show which of the brain areas showing *cfos* induction are directly activated by DT-BPNs in juveniles. We currently do not have the tools

to address this question functionally in juveniles. We have therefore narrowed our claim for the role of the circuit in the abstract and now state that it ‘drives social affiliation’.

However, we believe our larval EM data is also predictive of functional connectivity at the juvenile stage. As additional evidence for functional connectivity of putative larval BPN neurons with target areas, we have now specifically analyzed the presence of synaptic contacts.

We found that all BPNs we investigated, which make axonal projections to ‘social network’ brain regions, also made synaptic contacts in these regions, suggesting a functional link.

(28)

4) The authors could have done more to explain how they achieved imaging in juvenile zebrafish, a method that has long been thought elusive and too difficult. Is there any behavioral indication that the social behavioral patterns are still intact in the configuration under the microscope? How long is the preparation stable? Are there any differences in handling, recorded variables that indicate health? Why was no imaging performed in the ‘social network’? Please elaborate in the methods section exactly how and provide supplementary figures presenting the configuration.

We now added an extensive description in the methods and a new supplementary figure S3. In it, we also provide previously unpublished details on the implementation of the remote focusing module on the 2-photon microscope.

As pointed out in our response to comment 2 (reviewer 1) and comment 15 (reviewer 2) we focused our calcium imaging on retinorecipient areas that were accessible to volumetric imaging.

The current preparation represents a compromise between imaging stability for unambiguous segmentation of neuronal ROIs over long GCaMP recordings (>30 minutes) and health. Generally, juveniles were healthy for at least 60 minutes under the microscope and most animals readily resumed normal swimming upon release from the agarose. Survival depends on oxygenation and efficient osmoregulation, which requires freeing the mouth and tail from the agarose (see Bergmann et al. 2018, JDevBio). However, our preparation is not yet optimized for behavioral analysis and we cannot provide evidence that social behavioral patterns are intact under the microscope. This will be an important extension for future investigation. We expect that embedded behavior will be less robust than free-swimming, but largely intact, similar to our experience with hunting behavior in embedded larvae (e.g. Semmelhack et al. 2014. eLife).

In the meantime, we think the spatial correspondence of tectal and thalamic GCaMP activity patterns with *cfos* signal from free-swimming animals supports the notion that measurements in these areas from immobilized animals recapitulate the free-swimming condition.

Also see reply to comment # 16 (reviewer 2).

(29)

5) Please provide a supplementary movie/ imaging stack (data) of the functional imaging in juvenile fish.

We now added a supplementary movie S4.

(30)

6) It would help if the authors discussed how the BPN selectivity is created and how their responses may serve to separate behaviors. What is the relationship between TeO and DT BPNs?

Given the constraint on word count, we do not have an opportunity to discuss these important but more speculative points in detail in the main text.

-We currently do not know how BPN selectivity is created computationally. We can only predict the existence of a temporal band-pass filter in the detection of local acceleration at or upstream of BPNs.

-As the reviewer suggests in their conceptual concern #2, we find it attractive to speculate that movement patterns could serve as a species and/or stage specific recognition signal. In addition, BPNs should also be effective for detecting conspecifics against coherent motion of debris particles in the water.

-Based on the necessary role of the zebrafish TeO in target directed hunting and its retinotopy, we predict that tectal BPNs inherently encode the spatial location of BM stimuli. SAGFF(lf)81c ablations and our EM data suggests that DT-BPNs are driven by tectal input, in some cases converging from both brain hemispheres with no apparent retinotopy (data not shown). Our current hypothesis is that DT-BPNs might encode the attractive stimulus quality. DT ablations suggest that this signal is also required for shoaling. In addition, a hypothetical DT 'quality signal' may also mediate other systemic effects of social affiliation such as social buffering but this is speculation.

(31)

7) It would be great if they had expanded on the broader view in the literature on how social signals are encoded. As is, the reader learns almost nothing about how these results can be interpreted in the larger field of social cue processing.

Due to the word limit, we would like to point to the references covering this fascinating field that we included in the main text.

(32)

8) Another limitation is that the authors focused only on the BPNs without probing or discussing the relevance of the downstream processing and how the social attraction could be implemented. It would be better if this were discussed in more detail in both the main text and the discussion. And in the future, it would be great to see if all neurons with shared anatomical characteristics with BPNs process social information. This team recently built a fantastic technique described in Kramer et al. 2019- Function-guided inducible morphological analysis (FuGIMA). This could be used to elucidate whether actual, not putative BPN projection patterns and compared to the EM data.

Given the word limit, we do not have an opportunity to discuss these points which were not explicitly experimentally assessed. We would like to stress that our manuscript describes the discovery of neuronal sensory circuitry that detects BM during shoaling. We agree that it will be interesting in the future to reveal how this signal is used by downstream circuits to guide motor decisions for steering to bring about social attraction.

(33)

9) Given the presented data in Figure 2, why was there no imaging performed in tectal ablated 21-day old fish (only behavior, Figure 4)?

Thank you for encouraging the collection of these additional data. We have now repeated the tectal ablation experiment both in 21-day old fish and additional 6 dpf fish. We replaced the data in Figure 4 to show the juvenile data set and added a supplementary figure S8 with the combined larval data. The severe reduction of thalamic responses to moving dots is qualitatively very similar between the stages. Additionally, we found a stronger reduction in thalamic and pretectal responses to control stimuli in juveniles compared to larvae.

(34)

10) Finally, a cell-type-specific model that would illustrate the predictions for perturbation experiments would be much appreciated. Currently, the understanding of the circuit computation appears to be limited to the mere existence of BPNs in the TeO or DT. Understanding what functional role they play likely requires a model or clear hypothesis of how their activity leads to social attraction or repulsion.

Our manuscript describes the discovery of socially activated BPNs in TeO and DT, their requirement for affiliation (shoaling) and EM level connectivity. We have not experimentally or analytically assessed, nor do we make claims about more specific computations, or how such computations change over development from repulsion to attraction.

(35)

If these points would be addressed, new data demonstrate causality in this circuit including BPNs, and whether/how this circuit can lead to repulsion in larva and attraction in juveniles, the presented research could reach a very high impact level, justifying the publication in this journal. These insights into how vertebrate brains can use visual signals as social cues to direct behavior are very important for our understanding of complex neural processing.

Thank you for your appreciation of the importance of these insights. We think our DT ablations provide strong causality for BPNs during shoaling.

More research is necessary to test the hypothesis that this circuit controls the transition from repulsion to attraction. Alternatively, the BPN circuit may only trigger attraction upon maturation in juveniles, and a separate circuit may trigger repulsion differentially during development. Recent behavioral analyses of larval repulsion and its transition into juvenile shoaling offered complementary, and potentially conflicting predictions for the underlying neural substrates. (Groneberg et al 2020 Current Biology) highlight a primary role for

mechanosensation in larval repulsion while (Harpaz et al 2021 Nature Communications) highlight similar visual (behavioral) algorithms and a sign switch during maturation from larvae to juveniles.

In the future, it will be exciting to study circuit activity associated with behavioral readouts of shoaling across the full network and across the development of shoaling to identify the neuronal implementation of this behavioral switch.

Reviewer Reports on the First Revision:

Referees' comments:

Referee #2 (Remarks to the Author):

The authors have addressed most of my previous concerns. The most important revision is the addition of functional manipulations using the s1026Et driver and laser ablations of DT. These strengthen the paper substantially. The apparently quite specific effects strongly support the conclusion that the PVPN-DT axis is a core component of a network controlling behavioral responses to stimuli mimicking social visual input. The conclusions with respect to social signals and behavior remain a bit indirect because the most important results were obtained with moving dot stimuli, rather than real social input. Nevertheless, the results are of high interest.

Comments on the manuscript:

The fraction of neurons classified as BPNs in larvae was only slightly lower than in juveniles. Can this be accounted for by differences in the field of view? Was it statistically significant? Was the fraction of neurons in DT (and other nuclei) lower (or higher) than in juveniles?

Minor comments:

“Virtual conspecifics activated a cluster in the dorsal thalamus (DT), while real conspecifics additionally activated an anterior cluster in the ventral...”: misleading: real conspecifics also activated DT.

Reporting the response of DT-BPNs to motion stimuli as a function of speed is ambiguous without additional information about the size and distance of a visual stimulus of a given size from the fish. Please include this information. Otherwise, report the angular velocity and size.

Fig S7: please explain MTZ and RFP in the legend for f,g.

Referee #3 (Remarks to the Author):

The authors thoroughly addressed all my concerns, and thoughtfully responded and fixed issues wherever possible or explained their reasoning why certain requests are beyond the scope of this manuscript. However, while the ablations are a great step towards establishing causality, these experiments alone do not truly satisfy the need for a mechanistic description of the function of the neural network driving social affiliation. Altogether the manuscript is much improved and will be of high interest to related fields.

Two comments:

Why is there no code deposited in the bit bucket link?

Ideally, a sham ablation would have been added to the DT laser ablation.

Author Rebuttals to First Revision:

Response to reviewer comments

Referee #2 (Remarks to the Author):

The fraction of neurons classified as BPNs in larvae was only slightly lower than in juveniles. Can this be accounted for by differences in the field of view? Was it statistically significant? Was the fraction of neurons in DT (and other nuclei) lower (or higher) than in juveniles?

Generally, the anatomical sampling by volumetric 2-photon imaging varied between animals and age groups (see also figure S4a). For example, the fraction of neurons recorded in DT was higher in larvae (25% for larvae and 17% for juveniles), though not statistically significant. Because of sampling differences like this and ambiguity at the fine-level correspondence of DT boundaries (and other brain areas) between zebrafish stages, we refrained from quantitative comparisons of absolute and relative frequencies of neurons between age groups. Instead, we focus on the qualitative statement that DT-BPNs are also present in zebrafish larvae.

Minor comments:

“Virtual conspecifics activated a cluster in the dorsal thalamus (DT), while real conspecifics additionally activated an anterior cluster in the ventral...”: misleading: real conspecifics also activated DT.

We now explicitly state this as suggested by the reviewer.

Reporting the response of DT-BPNs to motion stimuli as a function of speed is ambiguous without additional information about the size and distance of a visual stimulus of a given size from the fish. Please include this information. Otherwise, report the angular velocity and size.

We added this information to the methods section where we describe visual stimulation for immobilized animals.

Fig S7: please explain MTZ and RFP in the legend for f,g.

We added this information to the legend.

Referee #3 (Remarks to the Author):

Why is there no code deposited in the bit bucket link?

The code is now available in the repository.

Ideally, a sham ablation would have been added to the DT laser ablation.

We agree in principle that a sham ablation can be a suitable control group. However, the concept of a sham ablation is perhaps better suited to study the differential requirement of distinct groups of neurons within one brain area, rather than assessing the overall involvement of an entire area. We initially attempted to ablate individual BPNs in DT. In this case it would have been a suitable sham control to ablate other DT neurons with a preference for continuous motion, for example. However, we did not succeed in ablating large numbers of DT-BPNs bilaterally in juveniles due to preparation-health constraints.

In the context of ablating a brain area, the purpose of a sham ablation is less clear. There could be ablation effects which might spread beyond the lesion site such as heating of the brain, blinding, bleeding, or secondary effects due to tissue repair responses. Specifically for DT, for example, we empirically found that we had to keenly avoid damage to a blood vessel passing nearby, or risk a major stroke. An off-target effect like this is highly region specific and cannot be controlled by a randomly placed sham ablation of another brain area. We thus argue that selecting another brain area for a

sham ablation is rather ambiguous, especially for a behavior that engages many brain areas. Instead, we focused on demonstrating specificity of the laser ablation to attraction versus repulsion, and showing overlap with the phenotype after chemogenetic ablation.